# GazeVLM: Gaze-Guided Vision-Language Models for Efficient and Robust Inference

## Abstract

Vision-language models (VLMs) are emerging as a core building block of modern intelligent assistants, enabling real-time human-machine interactions based on natural language and vision. However, the excessive number of visual tokens generated from images results in high latency, low throughput, and memory bottlenecks, which hinder real-time interactions in resource-constrained settings. To address this, we aim to reduce the number of tokens by prioritizing tokens for efficient inference using user-relevant context. With the growing usage of smart glasses, eye gaze has emerged as a promising sensing modality that can naturally convey the user intent and interests based on the user's viewing context. Therefore, it can provide useful hints for efficient inference. However, the robustness of gaze-aware VLM depends highly on the saliency of gaze data. When gaze data is inaccurate, the model may overlook informative visual content, leading to degraded inference accuracy. To this end, we introduce GazeVLM, a novel gaze-guided context-aware VLM framework for efficient and robust inference under a token budget constraint. GazeVLM consists of two key phases: (i) GazeVLM-Pre: a gaze-aware preprocessing mechanism before image encoding that extracts user-attentive scenes while not losing the global understanding for robust inference; (ii) GazeVLM-Post: a gaze-guided token selection method after image encoding that prioritizes tokens around the gazing area for efficient inference under the token budget constraint. Through extensive experiments using two visual question answering datasets with real human eye-tracking data, we demonstrate that GazeVLM achieves both efficiency and robustness under varying token budgets and gaze data qualities, outperforming diverse gaze-aware and gaze-agnostic baselines. Specifically, given the budget of 500 tokens ($\approx$22% of the tokens of the vanilla architecture), we can achieve up to $1.9\times$ higher throughput and 37% lower latency while slightly improving accuracy compared to the vanilla architecture.

## 1 Introduction

With the growing market of extended reality (XR) and smart glasses, artificial intelligence (AI) assistants are increasingly integrated into daily eyewear devices, such as Meta Aria Gen 1 and Gen 2 (Engel et al., 2023), or Meta Ray-Ban Glasses equipped with the Meta AI and built-in cameras (Waisberg et al., 2024; Meta Platforms, Inc., 2025). For example, a user wearing smart glasses may ask, "What does this sign mean?" while looking at a street sign. Then, the AI assistant is expected to be aware of the street sign to which the user is referring through the built-in camera and respond in real-time. This capability is empowered by vision-language models (VLMs), such as GPT-5 (OpenAI, 2025), Gemini (Team et al., 2023), LLaMA4 (Meta, 2024), and LLaVA (Liu et al., 2023a;b; 2024b), which enable natural language interactions grounded in the user's viewing context (Lee et al., 2025; Wang et al., 2024). However, supporting such applications requires meeting two key requirements: (i) they require *efficient inference* to enable real-time, interactive responses; (ii) they are expected to respond based on *user-relevant, context-aware* understanding.

The first requirement–efficient inference–remains a significant challenge for VLMs due to their computational overhead (Zhao et al., 2025; Yang et al., 2024). This is primarily constrained by processing the image input. In a typical VLM inference pipeline, the image is processed by a vision encoder such as vision transformers (ViTs) (Dosovitskiy et al., 2020), which generates a set of visual tokens. These visual tokens (from the input image) with textual tokens (from the user prompt) are

fed into a large language model (LLM) to generate the response (see more details in Sec. 2.1). Here, while the number of textual tokens is proportional to the length of the user prompt, generally on the order of tens, the number of visual tokens is often much larger, from hundreds to thousands.[1] This excessive number of visual tokens results in inefficient inference in terms of high latency, low throughput, and memory bottlenecks. To address this issue, (visual) token selection has become a promising approach for efficient VLM inference (Arif et al., 2025; He et al., 2024; Shang et al., 2024; Zhang et al., 2024b). However, aggressively removing tokens risks missing important tokens and can harm accuracy. This trade-off raises an intrinsic question: *How to identify informative tokens within a constrained budget without sacrificing accuracy?*

The second requirement–user-relevant, context-aware understanding–can be a useful hint for answering the above question and guiding token selection. Among the modalities that provide user context, *eye gaze* intuitively reflects user intent–what and where the user is referring to and interested in (Zhang et al., 2024a; Wilson et al., 2025). As the computation bottleneck in VLMs is often driven by the visual content that becomes tokens, the gaze data–also associated with the same visual content–offers a promising means of identifying important visual tokens. Recent works have also explored visual token selection approaches for efficient inference (Arif et al., 2025; Shang et al., 2024). However, they prioritize visually important tokens based on model-derived metrics such as ViT attention scores, which are not necessarily aligned with the user's intent. In stark contrast, we prioritize tokens via gaze guidance, which directly reflects the user intent within the user's viewing image context. Accordingly, the gaze data is leveraged in two image-associated stages of the inference pipeline: before and after image encoding.

Very few existing works attempt to incorporate eye gaze into VLMs, with exceptions including GazeLLM (Rekimoto, 2025) and GazeGPT (Konrad et al., 2024), which leverage eye gaze to crop raw images to focus on the gazing area. However, these gaze-aware VLM methods have two critical limitations. First, since they only consider image preprocessing before feeding the image into the VLM, *they fail to meet the token budget for efficient inference*. Image encoders in VLMs can only accept and produce a fixed number of tokens regardless of the image size, and thus, preprocessing cannot enforce budget constraints (see detailed pipeline in Sec. 2.1). Second, *these approaches crop the image and discard surrounding areas from the very beginning, thereby losing important global information*. This incurs a severe problem when the saliency of gaze data is low; that is, important content can be mistakenly removed (see Sec. 3). Therefore, one needs to consider both *efficiency* under different budget constraints and *robustness* under various gaze data qualities.

To ensure both efficiency and robustness while integrating gaze data, we draw inspiration from foveated rendering (Albert et al., 2017; Patney et al., 2016), where the foveal area (centered around the gaze point) is rendered at a higher resolution, while the peripheral areas have a lower resolution. Foveated rendering enables computation-efficient, low-latency rendering without degrading the user experience. Inspired by this idea, we adopt a similar approach for VLM inference by downweighting less-attentive areas without fully excluding them, allowing the model to maintain a global view of the scene for robustness while focusing more on the gazing area for efficiency.

To this end, we propose GAZEVLM, an efficient and robust VLM inference framework via user-centric gaze guidance under a token budget constraint. As shown in Fig. 1, GAZEVLM consists of two phases. First, we design GAZEVLM-PRE, a gaze-aware image preprocessing phase before encoding for robust inference. Specifically, we adopt two complementary image views: a local-view image focused on the user-attentive area and a global-view image for global understanding. Gaze data is used to extract the narrowed gazing area for the local-view

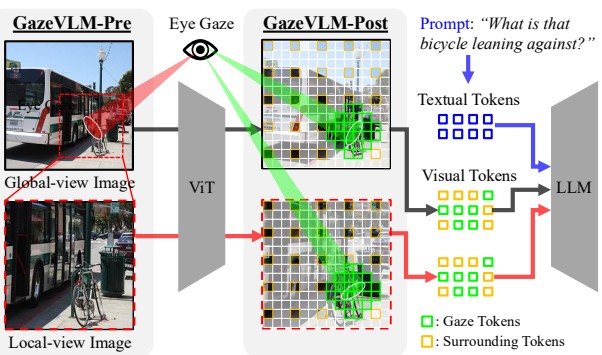

Figure 1: Overview of GAZEVLM.

[1]For example, a single image in LLaVA-Next (Liu et al., 2023a; 2024b) corresponds to 2,880 tokens and takes nearly 20 seconds to complete a single inference (Arif et al., 2025).

image. However, this local-view image may become less relevant when the gaze data is inaccurate. To compensate for this, our global-view image captures the entire scene to maintain global coverage and ensure robustness across various gaze data qualities. Second, to satisfy the token budgets, we introduce GAZEVLM-POST, a gaze-guided token selection phase for efficient inference. Along with the gaze-aware tokens, we also uniformly sample tokens across the whole tokens to retain the global context. By combining gaze-aware preprocessing with token selection, we can achieve a focused yet holistic understanding of the user's viewing context and thus ensure both efficiency and robustness.

To the best of our knowledge, *this is the first work that integrates eye gaze for efficient and robust VLM inference*. Our main contributions can be summarized as follows:

(i) We introduce GAZEVLM, the first gaze-guided token selection framework for efficient VLM inference. By prioritizing tokens around the gazing area, GAZEVLM improves throughput, latency, and memory usage. Unlike previous efficient VLM methods that prioritize visually important content, which may not align with user intent, our gaze-guided GAZEVLM directly selects tokens of what the user is actually attending to, and thus offers context-aware efficient inference.

(ii) We also design a gaze-aware preprocessing mechanism for robust VLM inference. Existing gaze-aware VLMs show accuracy degradation under low-saliency gaze by discarding the surrounding areas. GAZEVLM, however, is designed to balance both local and global context. This ensures robust inference without sacrificing accuracy across varying gaze data qualities.

(iii) We conduct extensive experiments on two visual question answering (VQA) datasets using real eye-tracking data.[2] Comparing with various gaze-aware and gaze-agnostic baselines, we demonstrate the efficiency and robustness of GAZEVLM across various token budgets and gaze qualities. Specifically, GAZEVLM can achieve up to $1.9\times$ higher throughput and 37% lower latency compared to the vanilla architecture while using only 22% of the tokens.

## 2 BACKGROUND

### 2.1 INFERENCE PIPELINE OF VLMS

Recent VLM architectures (e.g., LLaVA-Next (Liu et al., 2023a; 2024b)) consist of an image encoder and an LLM. The inference pipeline includes: (i) image preprocessing via resizing or partitioning; (ii) image encoding using the vision encoder such as ViTs; (iii) projecting the image embeddings into the language space; and (iv) generating the response by feeding all tokens (including the visual tokens from the image and the textual tokens from the prompt) into the LLM.

Commonly used image encoders, such as ViTs, extract visual features at the patch level, where each patch is a block of image pixels. These encoders process only a fixed size of images, e.g., ViT-L/14@336 always resizes any input image to a single low-resolution image of $336\times336$ pixels, regardless of the image size. The image is then encoded and generates 576 visual tokens. As this results in losing the details and thus degrading the accuracy, recent works adopt dynamic partitioning by dividing the image into multiple fixed-size partitions (Chen et al., 2024a; Liu et al., 2023a; 2024b; Wu et al., 2024). For example, as illustrated in Fig. 6 (see Appendix A.2), rather than a single $336\times336$ pixel image, LLaVA-Next uses five images of $336\times336$ pixels (from the image resized to $672\times672$ pixels): one from the resized full image, and the other four from the input image being uniformly divided into $2\times2$ grids. While multiple partitions can improve accuracy by supporting high resolutions, they proportionally increase the number of visual tokens and incur inefficient inference.

**Possible stages for efficient inference.** We consider two hook points to incorporate eye gaze for efficient inference: (i) controlling the number of partitions from the input image *before image encoding*; and (ii) controlling the number of tokens from each partition *after image encoding*. While it is also viable to drop tokens within LLMs (Chen et al., 2024b; Huang et al., 2025), such methods still require loading all tokens first and thus cannot satisfy the token budget; therefore, we focus on the first two stages before and after the image encoding. At *the first hook point before image encoding*, we can reduce tokens by controlling the number of image partitions. Fewer partitions lead to fewer tokens, which accelerates inference but may harm accuracy. The limitation is that the number of visual tokens is exactly proportional to the number of partitions, and thus, token control can only

---

[2]AiR-D and VQA-MHUG are the only publicly available VQA datasets with real human eye-tracking data.

Table 1: Performance of gaze-aware VLMs under different gaze qualities.

| Approach | # Visual Token | Throughput (token/s) | Accuracy | | |
|---|---|---|---|---|---|
| | | | High-Saliency | Low-Saliency | All |
| Vanilla | 2,295 | 1.42 | 74.8 | 74.2 | 74.3 |
| Dotted Map | 2,295 | 1.42 | 76.7 (+1.9) | 73.3 (-0.9) | 74.0 |
| Circled Map | 2,295 | 1.41 | 78.8 (+4.0) | 73.6 (-0.6) | 74.7 |
| Gaussian Blur | 2,295 | 1.25 | 75.4 (+0.6) | 71.6 (-2.6) | 72.4 |
| Heatmap (Voila-A) | 2,295 | 1.33 | 76.8 (+2.0) | 71.4 (-2.8) | 72.5 |
| Cropping (GazeLLM) | 2,926 | 1.27 | 76.4 (+1.6) | 71.6 (-2.6) | 72.6 |
| Cropping (GazeGPT) | 1,800 | 1.62 | 76.1 (+1.3) | 71.8 (-2.4) | 72.7 |

be applied at the partition level (e.g., always multiples of 576 tokens in ViT-L/14@336). *The second hook point is after encoding but before the LLM*. We can freely drop tokens regardless of partitions, which allows for fine-grained token budget control.

## 2.2 RELATED WORK

**Efficient VLMs.** Using model-derived metrics such as attention scores, several works on efficient VLMs select or compress tokens after image encoding but before LLMs (Arif et al., 2025; He et al., 2024; Shang et al., 2024; Zhang et al., 2024b; Yang et al., 2024) or during decoding in LLMs (Chen et al., 2024b; Huang et al., 2025; Liu et al., 2024a; Lin et al., 2025). While they effectively reduce the number of tokens and thereby improve the throughput or memory usage, the generated tokens are computed over the entire visual patches. That is, they may have already encoded distracting information from irrelevant or uninformative regions, similar to long-context LLMs, where many less important tokens can dilute understanding, making it difficult to attend to key visual content. More importantly, they often focus only on visually important features, which are not necessarily relevant to user intent. Our approach is fundamentally different: we directly incorporate gaze data to guide token selection that is better aligned with the user's interest.

**Gaze-integrated VLMs.** Recent works have incorporated gaze data into VLMs (Konrad et al., 2024; Rekimoto, 2025; Yan et al., 2024). *Voila-A* overlays a Gaussian heatmap of eye gaze onto the original image and integrates eye gaze data into an attention mechanism for better understanding (Yan et al., 2024). However, it requires fine-tuning/retraining, while we provide a plug-and-play manner. For gaze-aware VLM inference approaches without additional training, *GazeLLM* crops the video frame based on the gaze point (Rekimoto, 2025). Similarly, *GazeGPT* crops the image into multiple partitions with different granularity (Konrad et al., 2024). However, cropping fully excludes surrounding areas, which has a risk of losing valuable visual content. When the gaze data is less accurate, cropping may lose visually important objects when the gaze data is less accurate. Also, if the gazing object is larger than the cropped area, this object cannot be fully captured. Therefore, all of gaze-aware VLM methods are sensitive to gaze qualities. Moreover, as these methods are processed before the image encoding, they cannot control the token budget.

*To the best of our knowledge, no existing work has leveraged the gaze data to select tokens for efficient VLM inference*, as summarized in Table 6 (see Appendix Sec. A.2.1).

## 3 KEY INSIGHTS

We implement gaze-aware VLM methods using the AiR-D dataset (see details on datasets and baselines in Sec. 5). We measure the efficiency in terms of (i) the number of visual tokens and (ii) throughput in terms of the number of generated tokens per second. We also evaluate the accuracy in three groups based on gaze saliency: (i) all samples; (ii) a high-saliency group of the top 10% samples with low deviations from the centroid; and (iii) a low-saliency group consisting of the rest. We further examine the results under varying group splits and criteria in Fig. 14 (see Appendix A.3.2).

**Efficiency for constrained token budgets.** As shown in Table 1, there is no difference in the number of tokens except cropping methods. As all methods except cropping simply preprocess the

input image, the number of patches and thus the number of visual tokens remain the same from the model perspective. In contrast, cropping resizes the input image size, directly affecting the number of patches and thereby throughput. Specifically, *Cropping (GazeGPT)* has three partitions, resulting in fewer tokens than the other methods with five partitions. However, such preprocessing can only adjust the number of partitions and does not offer fine-grained token budget control. As described in Sec. 2.1, token budget can only be ensured after the encoding. Therefore, efficient inference requires selecting important tokens under a budget constraint after encoding. Here, eye gaze can serve as a strong indicator of user intent (Kumar et al., 2025; Sendhilnathan et al., 2024; Zhang et al., 2020). Evidenced by Table 1, all methods benefit from gaze integration when the gaze is less deviated, implying that the gaze is a useful cue for token selection in context-aware inference.

**Insight 1** *Efficiency can be ensured only in the postprocessing stage via token dropping. To select the tokens, not all visual tokens are equally informative; gaze-guided token selection can more intuitively reflect user intent than visual importance, leading to more relevant and efficient inference.*

**Robustness across gaze qualities.** Although the eye gaze can help improve inference efficiency, robustness is a natural challenge in using gaze data. As shown in Table 1, different from the high-saliency group, accuracy in the low-saliency group degrades, since inaccurate gaze points may lead to missing the actual user-interested area or object. Specifically, *Heatmap (Voila-A)* fails, as the overlay distorts the original color information. Cropping approaches or *Gaussian Blur* are particularly risky, as once informative visual content is lost from the beginning, it cannot be recovered. In contrast, *Circled Map* and *Dotted Map* only highlight the attentive area without fully discarding the surroundings areas, thus leading to a smaller accuracy drop. These findings suggest that integrating global understanding with focused attention enhances robustness across various gaze data qualities.

Furthermore, in VLM architecture, ViT divides the image into a fixed number of uniformly sized partitions regardless of the context or user interest (see Sec. 2.1). Then, each partition is encoded separately and given equal importance. As a result, user-attentive objects may be fragmented across multiple partitions, making it difficult for ViT to capture a complete understanding of the gazing objects. This context-agnostic partitioning often contains a partition that neither fully contains the objects of interest nor aligns with user intent. In contrast, by centering the input image around the user's gaze, we can ensure that the encoded region fully covers the user-attentive objects.

**Insight 2** *While eye gaze is helpful to highlight user-interested areas, low-saliency gaze data can misguide the model by removing informative areas. Thus, gaze-aware VLMs require the global context for robustness inference.*

## 4   OUR DESIGN: GAZEVLM

Given a token budget, denoted by $B$, we aim to maximize the inference accuracy within the budget. While improving accuracy is preferable, our main constraint is not to harm the accuracy, even with the reduced number of tokens. As depicted in Fig. 1, GAZEVLM consists of two phases: (i) GAZEVLM-PRE before encoding and (ii) GAZEVLM-POST after encoding.

**GAZEVLM-PRE: Gaze-Aware Preprocessing.** We first extract an image to emphasize a gaze-aware view. We prioritize the region where the user is looking at, while ensuring the user-attentive objects are fully captured. As discussed in Insight 2, although gaze is a key user-relevant factor, cropping methods that completely remove the surroundings risk losing the informative region (Konrad et al., 2024; Rekimoto, 2025). Therefore, retaining some surrounding information can provide the context and ensure robustness under various gaze data or images. Accordingly, as depicted in Tables 4 and 5 (see Appendix A.3), we extract two complementary image views from the original input image: a local-view image and a global-view image. Then, the gazing area has more visual tokens while each partition itself is self-contained without relying on other partitions.

The *local-view image* is a gaze-aware cropped partition of the global image, scaled by a ratio of $\alpha$, i.e., $P_l = \alpha \times P_g$, where $P_l$ and $P_g$ denote the pixels for the local-view and the global-view images, respectively. This local-view image allows the model to focus on the region relevant to the user's attention, capturing more fine-grained details for concentrated understanding. In contrast, the *global-view image* captures the full scene. It can provide a broad understanding of the entire scene to avoid overly relying on narrow regions, resilient across varying resource budgets, by compensating

for a poorly chosen crop, e.g., due to inaccurate gaze data, that could miss essential visual information. This *balanced gaze-aware selection with broad spatial coverage* can ensure robustness across varying gaze, image data, or budgets by maintaining global understanding (Shang et al., 2024).

The gaze region ratio $\alpha \in [0, 1]$ controls the size of the local-view image: a larger $\alpha$ indicates a wider view, while a smaller $\alpha$ leads to a more focused view on the gazing area. Based on our empirical results in Fig. 9a (see Appendix A.3), we set $\alpha = 0.5$, which reduces the local image to 25% of the global image. In some cases, the gaze point may be located near the edge of the image, and thus, we cannot extract a crop of size $P_l \times P_l$ centered at the gaze point. In such cases, rather than reducing $P_l$, we shift the crop window inside the image boundaries to ensure the crop size remains the same.

**GAZEVLM-POST: Gaze-Guided Token Selection.** Each of the two image partitions (local-view and global-view) is independently encoded, and $T_p$ tokens are generated for each image partition, totaling $2T_p$ visual tokens. Inspired by Insight 1, we integrate the gaze into a token selection to satisfy the token budget constraint. Let $T_a$ be the number of tokens to use. When the token budget $B$ is enough, i.e., $B \geq 2T_p$, we use all the tokens from both views without selection (i.e., $T_a = 2T_p$). However, when $B < 2T_p$, we only select a subset of tokens to fit into the budget (i.e., $T_a = B$). Let $T_v$ represent the token budget used for each view. To ensure fair coverage of both local-view and global-view images, we evenly allocate the token budget for each view (i.e., $T_v = \lfloor \frac{1}{2}T_a \rfloor$).

We not only select the tokens near the gaze point but also sample tokens from the surroundings to balance the concentration and coverage. Let $B_g$ and $B_s$ represent the budget for gaze tokens and surrounding tokens, respectively. For each view image, we distribute the token budget $T_v$ into tokens centered with the gaze point of allocated budget $B_g = \lfloor \beta \times T_v \rfloor$ and surrounding tokens with allocated budget $B_s = T_v - B_g$. *Gaze tokens* are selected based on the distance from the gaze area. We compute their Euclidean distances from the gaze point and choose the closest $B_g$ tokens in a circular region centered at the gaze point. *Surrounding tokens* are uniformly sampled across the remaining tokens, in the shape of grids, to get a holistic understanding spread over all the tokens.

Let $\beta \in [0, 1]$ denote the gaze token ratio. With $\beta = 1$, we select only gaze areas, while with $\beta = 0$, we uniformly sample tokens, regardless of the gaze. To balance local details and global coverage, we use $\beta = 0.5$, evidenced by our empirical results in Fig. 9b (see Appendix A.3). Note that selecting gaze tokens only (i.e., $\beta = 1$) is different from cropping, which loses surrounding information entirely. This is because token selection occurs after encoding, and token embeddings have already been computed over the whole input. Hence, gaze tokens alone still retain global understanding.

We detail the process of our token selection and provide examples of sampled tokens of GAZEVLM-POST in Algorithm 2 and Fig. 5, respectively (see Appendix A.1). We also present a VLM inference algorithm with GAZEVLM in Algorithm 1, along with the summary of notations in Table 3.

## 5 EXPERIMENTS

**Datasets.** We used two datasets. (i) AiR-D consists of 10,819 samples from 1,413 QAs by 20 participants (Chen et al., 2020). The QA sets are sourced from the GQA dataset (Hudson & Manning, 2019), while the gaze data is collected via the Vive Pro Eye headset. (ii) VQA-MHUG consists of 8,146 pairs, collected by 49 participants for 3,512 QA sets (Sood et al., 2021). The QA sets are selected from the VQAv2 validation set, and the gaze data is collected from the EyeLink 1000 Plus remote eye tracker. *To the best of our knowledge, these two are the only publicly available VQA datasets with real eye-tracking data.*

We used the last gaze point for each participant-QA pair. We further evaluated the performance under inaccurate gaze data and across various gaze extraction methods in Figs. 12 and 13, respectively (see Appendix A.3.2). Additional discussion is provided in Appendix A.4.

**Baselines.** As a baseline, (i) *Vanilla* is the original architecture with dynamic partitioning of five partitions. *Preprocessing baselines* process the input image before the encoder: (ii) *Full-Only*: using only the resized full image without any gaze guidance; (iii) *Dotted Map*: drawing a dot at the gaze point (Mani et al., 2020); (iv) *Circled Map*: drawing a circle centered at the gaze point; (v) *Gaussian Blur*: blurring the surroundings; (vi) *Heatmap (Voila-A)*: overlaying the Gaussian heatmap of the gaze point as in Voila-A (Yan et al., 2024); (vii) *Cropping (Center)*: cropping the center region of the image regardless of the gaze; (viii) *Cropping (GazeLLM)*: cropping a region centered at the gaze

Table 2: Performance of GAZEVLM against baselines. "N/A" means that the approach cannot control the token budget. The arrows indicate improvements compared to *Vanilla*.

| Approach | # Visual Token | | Latency (s) | | Memory | Throughput | Accuracy |
| | Budget | Actual | Total | Avg | (GB) | (token/s) | (%) |
|---|---|---|---|---|---|---|---|
| Vanilla | N/A | 2,295 | 8,358 | 0.77 | 14.7 | 1.42 | 74.3 |
| Full-Only | N/A | 576 | 6,122 | 0.57 | 14.3 | 2.51 | 71.5 |
| Dotted Map | N/A | 2,295 | 8,354 | 0.77 | 14.7 | 1.42 | 74.0 |
| Circled Map | N/A | 2,295 | 8,304 | 0.77 | 14.7 | 1.41 | 74.7 |
| Gaussian Blur | N/A | 2,295 | 9,231 | 0.85 | 14.7 | 1.25 | 72.4 |
| Heatmap (Voila-A) | N/A | 2,295 | 8,726 | 0.81 | 14.7 | 1.33 | 72.5 |
| Cropping (Center) | N/A | 2,928 | 9,623 | 0.89 | 14.9 | 1.19 | 55.5 |
| Cropping (GazeLLM) | N/A | 2,926 | 9,027 | 0.83 | 14.9 | 1.27 | 72.6 |
| Cropping (GazeGPT) | N/A | 1,800 | 7,771 | 0.72 | 14.6 | 1.62 | 72.7 |
| **GAZEVLM-PRE** | N/A | 1,200 | 6,751 | 0.62 | 14.4 | 2.02 (↑) | **76.4** (↑) |
| Spatial | 1,000 | | 6,371 | 0.59 | 14.4 | 2.05 | 72.2 |
| Spatial | 300 | | 5,748 | 0.53 | 14.4 | 2.35 | 65.9 |
| Sequential | 1,000 | | 6,490 | 0.60 | 14.4 | 1.98 | 63.2 |
| Sequential | 300 | | 6,073 | 0.56 | 14.4 | 2.16 | 46.7 |
| Pooling (2×2) | ≈25% | 583 | 5,721 | 0.53 | 14.4 | 2.39 | 71.1 |
| Pooling (4×4) | ≈7% | 150 | 5,476 | 0.51 | 14.4 | 2.37 | 61.1 |
| HiRED | 50% | 1,072 | 4,138 | 0.38 | 14.4 | 3.78 | 61.4 |
| HiRED | 15% | 314 | 5,780 | 0.53 | 14.3 | 3.78 | 61.1 |
| **GAZEVLM-POST** | 1,200 | | 6,711 | 0.62 | 14.4 | 1.91 (↑) | 74.3 (−) |
| **GAZEVLM** | ≥ 1,200 | 1,200 | 6,751 | 0.62 | 14.4 | 2.02 (↑) | **76.4** (↑) |
| | 1,000 | | 6,242 | 0.58 | 14.4 | 2.18 (↑) | 75.2 (↑) |
| | 500 | | 5,274 | 0.49 | 14.3 | 2.70 (↑) | 74.8 (↑) |
| | 300 | | 4,878 | 0.45 | 14.2 | **3.03** (↑) | 73.0 (↓) |

point (Rekimoto, 2025); (ix) *Cropping (GazeGPT)*[3]: extracting three crops centered at the gaze point at different scales (Konrad et al., 2024). *Postprocessing baselines* are applied after image encoding: (x) *Spatial*: uniformly sampling tokens across the entire image; (xi) *Sequential*: selecting the first certain set of tokens based on positional embedding order; (xii) *Pooling*: applying 2D-pooling; (xiii) *HiRED*: selecting important tokens based on ViT attention scores. Note that no existing work on efficient VLMs leverages gaze guidance to select tokens in the postprocessing stage (Arif et al., 2025). We further provide a summary of the baselines and examples of preprocessed images in Table 6 and Fig. 7, respectively (see Appendix Sec. A.2.1).

**Metrics.** We evaluate the performance in terms of overall accuracy and the following resource metrics: (i) *the number of visual tokens* fed into LLMs; (ii) the number of generated tokens per second as *throughput*; (iii) *total latency* of running all inference instances for each dataset; (iv) *average inference latency (per instance)* of generating the full response (also affected by the length of response); and (v) peak GPU *memory usage*. All metrics are averaged across the full evaluation set. We further reported the average power consumption per inference in Table 10 (see Appendix A.3.1).

**Implementation.** All experiments are conducted on a single NVIDIA A100-80GB GPU. We also report resource metrics using less powerful GPUs, including the NVIDIA L40S, Tesla P40, and NVIDIA A2 (entry-level GPU for edge use cases (NVIDIA, 2025)) in Table 10 (see Appendix A.3.1). Note that the accuracy is consistent across GPUs; only resource factors (e.g., latency, throughput, and power consumption) are affected. As a default setting, we use a batch size of one, and the maximum number of output tokens is 30. We provide further details and additional results in Appendices A.2 and A.3, respectively.

We present our experimental results and aim to answer the following research questions.

---

[3] As *GazeGPT* does not provide how to crop the image, we leverage five different cases and average them.

**RQ1: Can our gaze-guided method provide efficient inference without sacrificing accuracy?**
We first evaluate the resource efficiency and accuracy of GAZEVLM by varying the number of visual tokens. The budgets of 1,200, 1,000, 500, and 300 tokens correspond to 52%, 44%, 22%, and 13%, respectively, compared to the number of tokens used in the *Vanilla*[4]. As shown in Table 2, GAZEVLM on AiR-D with Mistral-7B at a 500-token budget improves throughput by 1.9×, reduces latency by 36%, and saves memory usage by 0.4 GB without harming accuracy. As we used the batch size of one, the total GPU memory usage does not seem significant. However, this will be more efficient in large batch sizes; as shown in Table 7 and Fig. 8 (see Appendix A.3), GAZEVLM at a 500-token budget can reduce 6.6 GB and 13.2 GB with the batch size of 16 and 32, respectively. We also report the results across datasets and models in Table 8 (see Appendix A.3.1).

Taking a closer look with varying numbers of tokens in Fig. 2, there appears to be a trade-off between accuracy and throughput; using fewer tokens increases throughput but degrades accuracy. However, GAZEVLM maintains the accuracy using only 500 tokens (≈22%), while throughput increases up to 300 tokens (≈13%). This indicates that GAZEVLM effectively prioritizes the gaze area and helps the model to focus on informative visual content. After a certain point, the throughput saturates due to the LLM generation; reducing input tokens affects the prefill (to compute the key-value cache), while the decoding (to generate the response) is relatively unaffected.

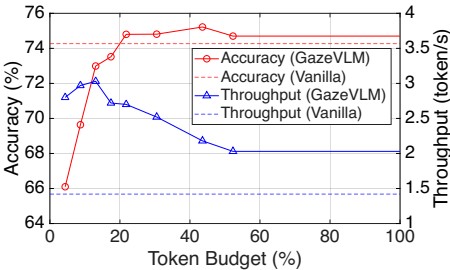

Figure 2: Accuracy and throughput across different token budgets.

**Comparison to baselines.** We compare GAZEVLM with several baselines across pre-encoding and post-encoding stages in Table 2. First, *Full-Only* is a single-partition baseline, which loses detail due to its low resolution. However, even with a single image, it has a whole understanding and thus can still achieve quite good accuracy, highlighting the importance of holistic understanding, which also aligns with our global-view partition. Several naive image preprocessing methodologies, including *Dotted Map*, *Circled Map*, *Gaussian Blur*, and *Heatmap (Voila-A)*, mask on top of the original image and thus cannot contribute to efficiency, as described in Sec. 3. We also compare GAZEVLM with three different cropping methods. Gaze-aware cropping approaches, e.g., *Cropping (GazeLLM)* or *Cropping (GazeGPT)*, can benefit from gaze data and thus perform better than gaze-unaware *Cropping (Center)*. However, cropping is generally sensitive to the crop size and cannot reliably maintain accuracy. These methods can provide good accuracy under some optimal crop sizes and good gaze qualities, but their performance highly fluctuates in our evaluation, depending on the crop size or gaze qualities. Furthermore, cropping methods cannot offer token-level budget control, as all cropped images are resized to a fixed size before encoding, always giving a fixed number of tokens. Although these methods can still control the token budget at a coarse level by reducing the number of partitions, resource efficiency has not been considered in any of these approaches.

In contrast, postprocessing baselines (e.g., *Spatial*, *Sequential*, and *Pooling*) can control the token budget by dropping or compressing tokens. These methods show stable performance with enough budget but drastically degrade with a tight budget. For example, with 300 tokens, *Spatial* and *Sequential* only achieve an accuracy of 65.9% and 46.7%, respectively. Even though they fail to provide a good performance, *Spatial* without any gaze data still outperforms *Sequential*, by uniformly sampling over the whole image, which implies that when the gaze data is missing or inaccurate, keeping the global information is promising, which motivates the need for surrounding tokens.

We also verify GAZEVLM against *HiRED*, a recently proposed attention-guided token dropping mechanism for efficient VLM inference. It performs poorly, even worse than *Spatial*, since it selects visually attentive tokens and captures main objects, rather than those relevant to the user's intent. This misalignment is particularly problematic in contextual VLMs where user attention, not visual importance, is tightly coupled with prompts. Thus, selecting user-focused tokens is essential. GAZEVLM outperforms all baselines, and the gap between ours and other resource-efficient methods becomes larger in more extreme cases. Specifically, with a budget of 300 tokens (i.e., 13% of

---

[4]LLaVA-Next includes padding tokens before encoding not to distort the input image ratio, which are then unpadded (i.e., dropped) after encoding. Therefore, the number of visual tokens injected into the LLM varies corresponding to the original image size, resulting in 2,295 tokens on average for the AiR-D dataset.

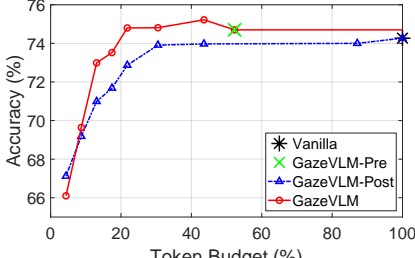

Figure 3: Ablation study

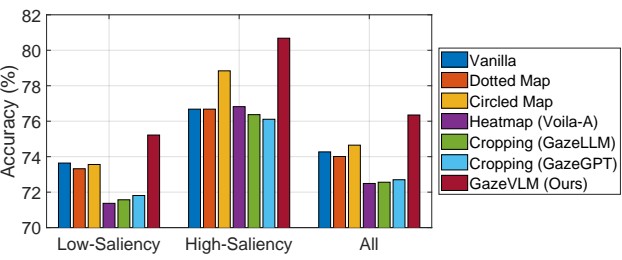

Figure 4: Robustness of GAZEVLM compared to gaze-aware baselines under various gaze qualities.

the tokens in the vanilla architecture), we can achieve an accuracy of 73.0%, while successfully providing high throughput and low latency, whereas all other efficient VLM methods can reach no more than 65.9%. This is achieved by balancing the gaze tokens with surrounding tokens (also evidenced by *Spatial* with reasonable accuracy) under flexible budget constraints.

**RQ2: Are both preprocessing and postprocessing essential?** To understand the importance of the two-phase processing in GAZEVLM, we conduct ablation studies by applying only preprocessing or postprocessing. As shown in Fig. 3, GAZEVLM-POST gradually degrades the accuracy as the token budget is limited and performs poorly compared to two-phase GAZEVLM due to its extensive information that may be irrelevant to the question and thus dilutes the understanding. On the other hand, GAZEVLM-PRE, which extracts gaze-aware regions before preprocessing, can achieve a good accuracy but lacks control budgets, similar to other preprocessing methods. In contrast, GAZEVLM with token dropping on top of preprocessed images consistently outperforms each stage of gaze-aware partitioning in terms of control flexibility and gaze-guided token dropping in terms of accuracy, offering a good trade-off between accuracy and resource control. Thus, GAZEVLM (with both GAZEVLM-PRE and GAZEVLM-POST) can achieve the budget controllability compared to GAZEVLM-PRE and better accuracy than GAZEVLM-POST. We also investigate whether two partitions in GAZEVLM-PRE are sufficient for inference in Fig. 10 (see Appendix A.3.1).

**RQ3: Can our gaze-guided concentration with global understanding ensure robustness?** We investigate accuracy across gaze qualities to assess the robustness of GAZEVLM. It should be noted that all baselines except GAZEVLM do not consider token efficiency and thus consume 1.5–2.4× more tokens, as explained in Table 2. The high-saliency group includes the top 10% of samples with low deviations, and the rest of the samples belong to the low-saliency group. As shown in Fig. 4, all approaches show accuracy improvement with high-saliency gaze. This means that integrating eye gaze contributes to improving the accuracy by highlighting the area that the user is interested in. Although *Circled Map* shows a slightly better accuracy than GAZEVLM in the high-saliency group, the actual difference is only 0.6% while consuming 1.9× tokens. More importantly, all baselines suffer an accuracy drop under low-saliency gaze and thus degrade the overall accuracy. This is because they either discard the informative visual content. In contrast, by balancing the global-view image with the gaze-focused area, GAZEVLM ensures robust inference even with low-saliency gaze. The impact of parameters for balanced design is further discussed in Fig. 9 (see Appendix A.3).

## 6 CONCLUSION

We introduced GAZEVLM, an efficient and robust inference framework for gaze-guided VLMs. We leverage the gaze-aware image preprocessing by extracting a gaze-centered local-view image along with a global-view image to ensure robustness. After image encoding, we further integrate gaze data into token selection by prioritizing the tokens near the gazing area, offering fine-grained token budget control. Based on our extensive experiments on two VQA datasets with real eye-tracking data, we demonstrated that GAZEVLM improves throughput, reduces latency, and saves memory usage while minimally affecting accuracy across token budgets and gaze data qualities.

Our work sheds light on the use of eye gaze for improving efficiency while ensuring robustness by integrating the user-relevant context into VLM inference. We believe that GAZEVLM can motivate contextual VLMs for intelligent multimodal assistants. For future work, we plan to create a VQA dataset of egocentric images/videos with synced eye gaze data. Moreover, we plan on implementing our framework on commercial devices (e.g., Meta Aria Glasses) to investigate its feasibility.

**Ethics Statement.** Our work utilizes human eye-tracking data, which can reveal user cognitive states, preferences, and attention patterns. Such data could be misused to manipulate user attention.

**Reproducibility.** Our code is publicly available at `https://github.com/anonymous-zzz/GazeVLM`, and the implementation details are described in Appendix A.2.

**LLM Declaration.** We used LLM to polish the paper and search for relevant terms. All technical content, experimental results, or analyses were solely provided by the authors.

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

# A APPENDIX

## A.1 ALGORITHMS

Table 3: The summary of notations.

| Stage | Notation | Description |
|-------|----------|-------------|
| GAZEVLM-PRE | $P_g$ | # of pixels in the width and the height of the global-view image |
| | $P_l$ | # of pixels in the width and the height of the local-view image |
| | $\alpha$ | Gaze region ratio |
| GAZEVLM-POST | $B$ | Token budget |
| | $T_a$ | # of tokens to actually use |
| | $T_p$ | # of tokens pf each image partition |
| | $B_g$ | Budget for gaze tokens |
| | $B_s$ | Budget for surrounding tokens |
| | $\beta$ | Gaze token ratio |

---

**Algorithm 1** VLM inference with GAZEVLM

---

1: **Input:** $\mathcal{I}_{\text{raw}}$: input image; $\mathcal{Q}$: query of user prompt with system template; $B$: token budget; and $\mathcal{F} = [x, y]$: gaze point; $\alpha = 0.5$: gaze region ratio; $\beta = 0.5$: gaze token ratio

2: Global-view image $\mathcal{I}_{\text{global}} :=$ Resized $\mathcal{I}_{\text{raw}}$ of $P_g \times P_g$ pixels;      ▷ GAZEVLM-PRE

3: $P_l \leftarrow \lfloor \alpha \times P_g \rfloor$;

4: Shift $[x, y]$ inside the image boundaries if needed;

5: Local-view image $\mathcal{I}_{\text{local}} := \mathcal{I}_{\text{global}} [x - \frac{P_l}{2} : x + \frac{P_l}{2}, y - \frac{P_l}{2} : y + \frac{P_l}{2}]$;

6: Visual token set of the global-view image $V_{\text{global}} = \textit{vision\_encoder}(\mathcal{I}_{\text{global}})$;

7: Visual token set of the local-view image $V_{\text{local}} = \textit{vision\_encoder}(\mathcal{I}_{\text{local}})$;

8: Token budget for each view $T_v = \lfloor \frac{1}{2} \times B \rfloor$;      ▷ GAZEVLM-POST

9: $B_g \leftarrow \lfloor \beta \times T_v \rfloor, B_s \leftarrow T_v - B_g$;

10: Tokens to inject into LLM $L = \textit{tokenizer}(\mathcal{Q})$;

11: **for** $V = [V_{\text{local}}, V_{\text{global}}]$ **do**

12:     **if** $length(V) < T_v$ **then**

13:        $L_{\text{gaze}} :=$ Gaze-guided $B_g$ tokens nearest from $[x, y]$;

14:        $L_{\text{surrounding}} :=$ Uniformly sampled $B_s$ tokens across the visual token set $V$;

15:        $L.append(sorted(L_{\text{gaze}} \cup L_{\text{surrounding}}))$;

16:     **else**

17:        $L.append(V)$;

18:     **end if**

19: **end for**

20: **return** response $\leftarrow \textit{language\_model}(L)$;

---

**Overall VLM inference with GAZEVLM.** The detailed procedure of the inference pipeline of VLM with GAZEVLMis described in Algorithm 1, and the notations are summarized in Table 3. Given an input image $\mathcal{I}_{\text{raw}}$ with a user prompt $\mathcal{Q}$, we aim to satisfy the budget for the number of visual tokens $B$, guided by the gaze point $\mathcal{F} = [x, y]$. In Line 2, we first extract the global-view image. In Lines 3–5, we extract the local-view image with the gaze region ratio of $\alpha$. Both view images are then processed via a vision encoder, as shown in Lines 6–7. In Line 8, the token budget is evenly divided between those two views. Among them, we allocate the budget $B_g$ for the gaze-centered with $\beta \times T_v$, and the rest for the surrounding tokens. In Lines 11–18, for each view, we select the $B_g$ tokens centered on the gaze point $[x, y]$, while $B_s$ tokens are uniformly sampled across the whole image. If the budget is enough, we do not drop any tokens, as shown in Lines 16–17. All the visual tokens are concatenated with the textual tokens in Line 10 and then injected into the LLM in Line 20.

**Token selection in GAZEVLM-POST.** In Algorithm 2, we provide the procedure of token selection in GAZEVLM-POST. It includes two stages: (i) gaze-token sampling (in Lines 3–6), where tokens

---

**Algorithm 2** Token selection in GAZEVLM-POST

---

1: **Input:** $\mathcal{F} = [x, y]$: gaze point; $B_g$: the budget for gaze tokens; $B_s$: the budget for surrounding tokens; $w$: the number of patches in width; $h$: the number of patches in height

2: Sampled points $sampled = []$;

3: $\mathcal{G} \leftarrow$ List of all $(i, j)$ points over the patches of $w \times h$;  ▷ Gaze token sampling
4: $dist \leftarrow euclidean\_distance(\mathcal{G}, \mathcal{F})$;
5: $\mathcal{G}' \leftarrow sorting(\mathcal{G}, dist)$; // sort $\mathcal{G}$ by $dist$ in ascending order;
6: $sampled.append(\mathcal{G}'[0 : B_g - 1])$;

7: $grid\_w, grid\_h \leftarrow w/\sqrt{B_s}, h/\sqrt{B_s}$;  ▷ Surrounding token sampling
8: **for** $w\_idx = 1 : \lfloor w/grid\_w \rfloor$ **do**
9:    **for** $h\_idx = 1 : \lfloor h/grid\_h \rfloor$ **do**
10:      **if** $len(sampled) > T_s$ **then**
11:       break;
12:      **end if**
13:      $sampled.append(\lfloor min(w-1, i \cdot grid\_w + \frac{grid\_w}{2}) \rfloor, \lfloor min(h-1, j \cdot grid\_h + \frac{grid\_h}{2}) \rfloor)$;
14:    **end for**
15: **end for**

16: **return** $sampled$;

---

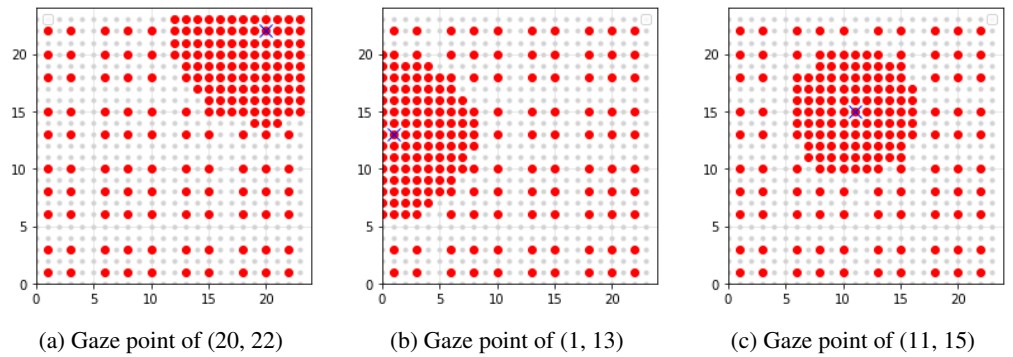

(a) Gaze point of (20, 22)       (b) Gaze point of (1, 13)       (c) Gaze point of (11, 15)

Figure 5: Examples to select 200 tokens (100 tokens each for gaze tokens and surrounding tokens) over 24×24 tokens.

are sorted by their Euclidean distance to the gaze point, and the closest $B_g$ tokens are selected; and (ii) surrounding-token sampling (in Lines 7–15), where the tokens are uniformly sampled over a grid until the budget is satisfied. This ensures to capture both gaze-focused details and coarse, uniform global context, as shown in Fig. 5.

### A.2 EVALUATION DETAILS

**Model configurations.** We have implemented GAZEVLM on top of `transformers` library (HuggingFace, https://github.com/huggingface/transformers). All parameters and constants are used as default settings. For example, in LLaVA-v1.6-mistral-7B configuration, the vision encoder has 24 hidden layers with the size of 1,024, an input image size of 336, an intermediate size of 4,096, 16 attention heads, and a patch size of 14 pixels, while the LLM is constructed from `mistralai/Mistral-7B-Instruct-v0.2` with an intermediate size of 14,336, 8 attention heads, and a vocabulary size of 32,064. The projection layer has a GELU activation function with a projection dimension of 768.

**Examples of our two-view images in GAZEVLM-PRE.** We show examples of global-view and local-view images from AiR-D and VQA-MHUG datasets in Tables 4 and 5, respectively. The global-view images are to capture a holistic understanding, while the local-view images provide a zoomed-in partition centered on the gaze point (marked with a yellow dot in each global-view

Table 4: Example of global-view and local-view images in the AiR-D dataset. The gaze point is marked with a red dot in the global-view image.

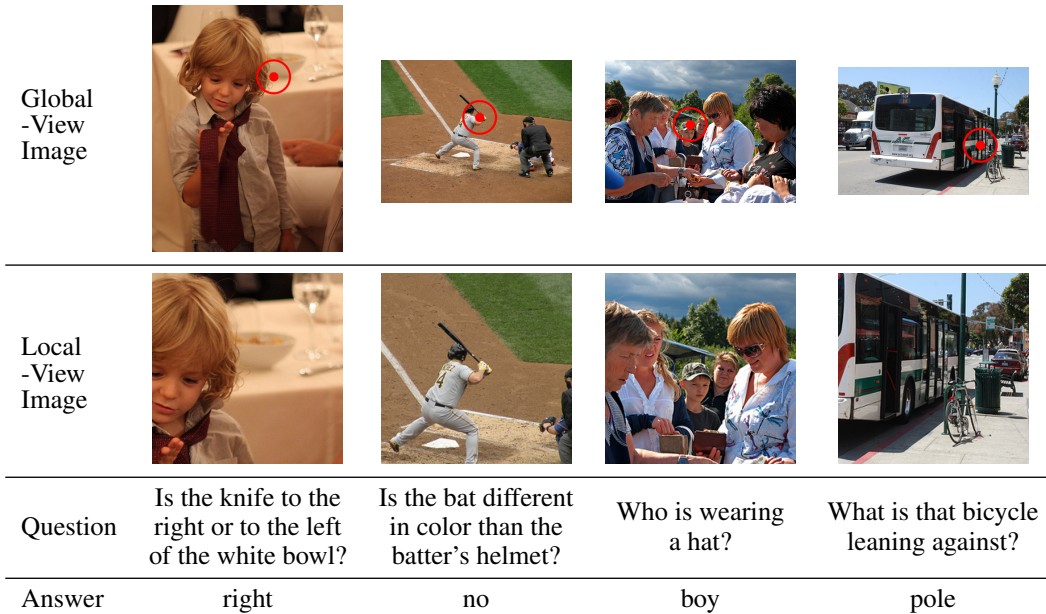

| | | | | |
|---|---|---|---|---|
| Question | Is the knife to the right or to the left of the white bowl? | Is the bat different in color than the batter's helmet? | Who is wearing a hat? | What is that bicycle leaning against? |
| Answer | right | no | boy | pole |

image). Local-view images can offer detailed and close-up visual content on specific regions of user interest, such as the last example of *What is that bicycle leaning against?* However, some questions– such as *Who is wearing a hat?*–may need broader understanding, highlighting the importance of global-view images.

### A.2.1 BASELINES

**Naive methods.** We illustrate the naive architectures of recent VLMs in Fig. 6.

First, *Full-Only* is an old architecture of LLaVA without dynamic partitioning, which is a previous version of LLaVA-Next. It resizes any input images into 336×336 pixels. The resized image is injected into ViT, and a certain number of visual tokens (e.g., 576 in ViT-L/14@336px) are generated. LLM processes visual tokens (from the image) and textual tokens (from the user prompt) and generates a response.

Second, *Vanilla* is an original architecture of LLaVA-Next with dynamic partitioning. As the old architecture of LLaVA, i.e., *Full-Only*, can only process one image resized into low-resolution of 336×336 pixels, it may lose details. To solve this problem, LLaVA-Next, i.e., *Vanilla*, leverages a dynamic partitioning. The dynamic partitioning has five partitions: one partition for the full image, same as *Full-Only*; and the other four partitions for detailed images. The input image is resized into a larger size of 672×672, and then the resized image is partitioned into 2×2 grids, generating four partitions. Five partitions are encoded separately, totaling five times more visual tokens than *Full-Only*, e.g., $576 \times 5 = 2,880$ tokens. These visual tokens are concatenated with textual tokens and then injected into the LLM.

**Summary of baselines.** We summarize baselines in terms of: (i) Gaze: whether gaze information is incorporated; (ii) Preprocessing: whether the method performs preprocessing stage before image encoding; (iii) Postprocessing: whether the method uses token selection after image encoding; and (iv) Budget Constraints: whether the method can satisfy the specified token budget constraints. As shown in Table 6, existing gaze-integrated VLM approaches only consider the preprocessing stage and thus cannot enforce token budgets. In contrast, GAZEVLM uniquely integrates gaze information into the inference pipeline under designated token budget constraints.

**Illustration of gaze integration approaches.** We illustrate the naive gaze-aware approaches in Fig. 7. *Dotted Map* (in Fig. 7a) and *Circled Map* (in Fig. 7b do not remove any visual con-

Table 5: Example of global-view and local-view images in the VQA-MHUG dataset. The gaze point is marked with a red dot in the global-view image. The VQA dataset of VQA-MHUG is VQAv2, which provides ten ground-truth answers.

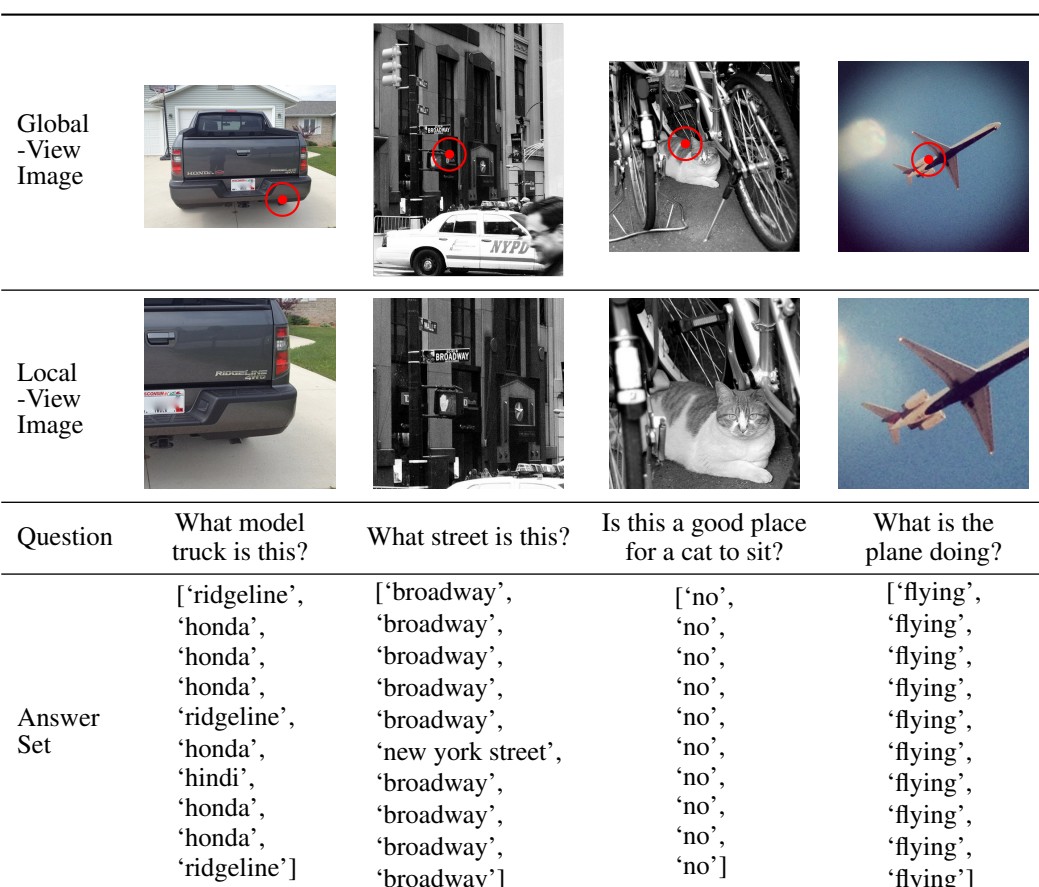

| Question | What model truck is this? | What street is this? | Is this a good place for a cat to sit? | What is the plane doing? |
|---|---|---|---|---|
| Answer Set | ['ridgeline', 'honda', 'honda', 'honda', 'ridgeline', 'honda', 'hindi', 'honda', 'honda', 'ridgeline'] | ['broadway', 'broadway', 'broadway', 'broadway', 'broadway', 'new york street', 'broadway', 'broadway', 'broadway', 'broadway'] | ['no', 'no', 'no', 'no', 'no', 'no', 'no', 'no', 'no', 'no'] | ['flying', 'flying', 'flying', 'flying', 'flying', 'flying', 'flying', 'flying', 'flying', 'flying'] |

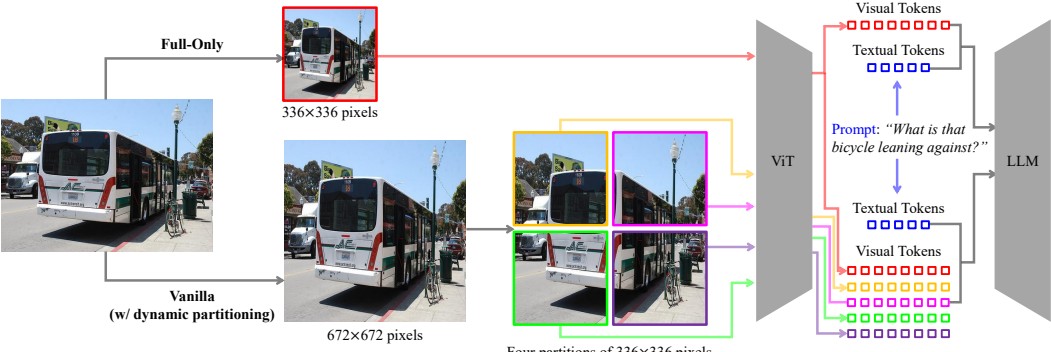

Figure 6: Architecture of *Full-Only* and *Vanilla*. *Vanilla* is an original architecture of LLaVA-Next, and *Full-Only* is LLaVA without dynamic partitioning, which is a previous version of LLaVA-Next.

tents but only highlight where the user is looking at. On the other hand, *Gaussian Blur* introduces smooth noise addition based on the distance from the gaze point and potentially loses visual information in blurred areas. Similarly, *Heatmap* applies a Gaussian heatmap with a JET colormap, and thus, there can be distortions by modifying the color.

Table 6: Summary of baselines.

| Approach | Gaze | Preprocessing | Postprocessing | Budget Constraints |
|---|---|---|---|---|
| Dotted Map | ✓ | ✓ | ✗ | ✗ |
| Circled Map | ✓ | ✓ | ✗ | ✗ |
| Gaussian Blur | ✓ | ✓ | ✗ | ✗ |
| Heatmap (Voila-A) | ✓ | ✓ | ✗ | ✗ |
| Cropping (Center) | ✗ | ✓ | ✗ | ✗ |
| Cropping (GazeLLM) | ✓ | ✓ | ✗ | ✗ |
| Cropping (GazeGPT) | ✓ | ✓ | ✗ | ✗ |
| Spatial | ✗ | ✗ | ✓ | ✓ |
| Sequential | ✗ | ✗ | ✓ | ✓ |
| Pooling | ✗ | ✗ | ✓ | ✗ |
| HiRED | ✗ | ✗ | ✓ | ✓ |
| GAZEVLM (Ours) | ✓ | ✓ | ✓ | ✓ |

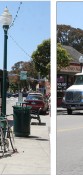 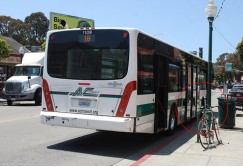 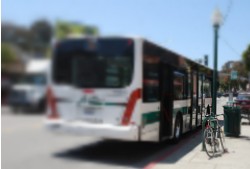 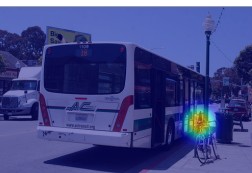

(a) Dotted Map      (b) Circled map      (c) Gaussian blur      (d) Heatmap

Figure 7: Examples of gaze-integrated VLM approaches.

Table 7: Memory usage and savings (GB) in budgets of 1,000 and 500 tokens with batch inference.

| Batch Size | 1 | 2 | 4 | 8 | 16 | 32 |
|---|---|---|---|---|---|---|
| Vanilla | 14.7 | 15.2 | 16.3 | 18.5 | 22.8 | 31.5 |
| GAZEVLM (1,000 tokens) | 14.4 (-0.3) | 14.6 (-0.6) | 15.1 (-1.2) | 16.2 (-2.3) | 18.2 (-4.6) | 22.3 (-9.2) |
| GAZEVLM (500 tokens) | 14.3 (-0.4) | 14.4 (-0.8) | 14.6 (-1.7) | 15.2 (-3.3) | 16.2 (-6.6) | 18.3 (-13.2) |

## A.3 ADDITIONAL RESULTS

**Batch inference.** To show the memory usage efficiency, we evaluate under different batch sizes. As shown in Table 7 and Fig. 8, although we only gain 0.3–0.4 GB memory usage reduction with the batch size of one, it linearly increases with the larger batch size. In typical VLM serving systems, they adopt multi-batch processing, and thus we can expect high memory reduction by dropping tokens, achieving savings by up to 9.2 GB and 13.2 GB when using 1,000 and 500 tokens, respectively.

**Parameter tuning.** We investigate the impact of different ratios on gaze-guided region or token selection from a relatively enough 1,000 token budget (44%) to a tight 300 token budget (13%). As shown in Fig. 9a, accuracy is affected by the gaze region ratio of the cropping size for local-view. Notably, too aggressive cropping (with low $\alpha$) can remove parts of the object or key visual information, leading to degraded accuracy. In contrast, a large region with a large $\alpha$ can be diluted without the benefit of utilizing gaze points, thus drastically harming the accuracy. The empirically optimal gaze region ratio is consistently around $0.3 \leq \alpha \leq 0.7$ across different budgets, implying that balancing local details and global coverage is essential.

Along with the balancing in GAZEVLM-PRE, we also investigate how to balance tokens between the gaze-centered area and the rest of the image in GAZEVLM-POST. A value of $\beta = 0$ means all tokens are uniformly sampled, while $\beta = 1$ selects only from the gaze-centered area. As shown

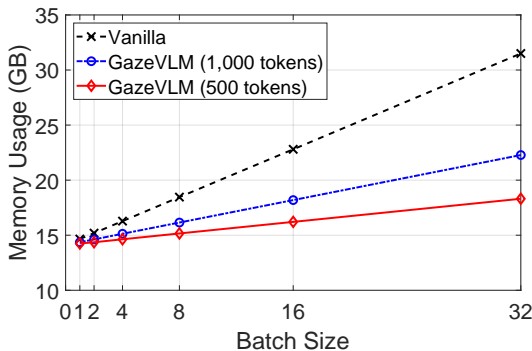

Figure 8: Memory usage with respect to the batch size.

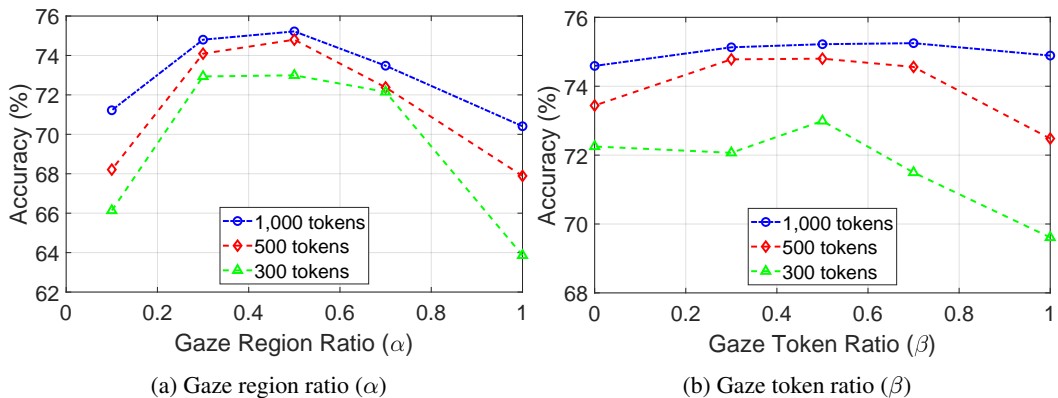

(a) Gaze region ratio ($\alpha$)   (b) Gaze token ratio ($\beta$)

Figure 9: Effect of different parameters.

in Fig. 9b, we find out that a balanced ratio (e.g., $0.3 \leq \beta \leq 0.7$) shows the best performance, highlighting the importance of combining gaze-guided tokens along with higher-coverage tokens. While we can achieve similar performance in the relaxed budgets, it becomes more sensitive under the limited budget, as tighter budgets lead to a smaller size of gaze-centered area. Note that while the local-view image in GAZEVLM-PRE is to exclude the non-gaze area in local-view, the gaze tokens serve as the anchor points, which still attend to the whole regions via self-attention during encoding (although they are spatially localized).

### A.3.1 GENERALIZATION STUDIES

**Different models and datasets.** We validate GAZEVLM across different datasets and models. As depicted in Table 8, LLaVA-Mistral-7B shows a stable accuracy even with 500 tokens in the AiR-D dataset and 300 tokens in the VQA-MHUG dataset, respectively. Interestingly, on the VQA-MHUG dataset, accuracy improves by 0.6%, 0.7%, and 3.2% using 1,200, 1,000, and 500 tokens, respectively. This may seem counterintuitive, as accuracy increases even under a more constrained token budget. However, we attribute this to a long-context LLM effect, as we described in RQ 2; rather than having more tokens, if we can select the most informative tokens effectively, fewer tokens can sometimes achieve better accuracy than a larger but less informative tokens.

For the different model of LLaVA-Llama3-8B, it shows a slight accuracy degradation on both datasets; using 1,000 tokens ($\approx$44%), accuracy shows a reduction of 2.7% and 1.0% for AiR-D and VQA-MHUG datasets, respectively. The reason is that LLaVA-Llama3-8B has a better LLM (i.e., larger model size and higher accuracy), and one of the goals of Llama3-8B is to mitigate a long-context impact, implying that it will be less affected by having more tokens. Therefore, even though there are fewer less-informative visual tokens, the LLM can differentiate between visual tokens that are more or less important regarding the prompt. Therefore, having more tokens does not degrade the accuracy. Importantly, even under such conditions, however, GAZEVLM provides

Table 8: Performance of GAZEVLM under various token budgets across models and datasets.

| Dataset | Model | # Visual Token | | Latency (s) | | Memory | Throughput | Accuracy |
| | | Budget | Actual | Total | Avg | (GB) | (token/s) | (%) |
|---|---|---|---|---|---|---|---|---|
| AiR-D | LLaVA-Mistral-7B | N/A | 2,295 | 8,358 | 0.77 | 14.7 | 1.42 | 74.3 |
| | + GAZEVLM | ≥1,200 | 1,200 | 6,751 | 0.62 | 14.4 | 2.02 (1.4×) | 76.4 (↑) |
| | | | 1,000 | 6,242 | 0.58 | 14.4 | 2.18 (1.5×) | 75.2 (↑) |
| | | | 500 | 5,274 | 0.49 | 14.3 | 2.70 (1.9×) | 74.8 (↑) |
| | LLaVA-Llama3-8B | N/A | 2,295 | 8,033 | 0.74 | 16.2 | 1.59 | 78.5 |
| | + GAZEVLM | ≥1,200 | 1,200 | 6,868 | 0.64 | 15.9 | 2.01 (1.3×) | 76.5 |
| | | | 1,000 | 6,494 | 0.60 | 15.8 | 2.16 (1.4×) | 75.8 |
| | | | 500 | 5,817 | 0.54 | 15.7 | 2.59 (1.6×) | 73.3 |
| VQA-MHUG | LLaVA-Mistral-7B | N/A | 2,278 | 7,641 | 0.94 | 14.7 | 1.16 | 66.8 |
| | + GAZEVLM | ≥1,200 | 1,200 | 6,667 | 0.82 | 14.4 | 1.37 (1.2×) | 67.4 (↑) |
| | | | 1,000 | 6,215 | 0.76 | 14.4 | 1.49 (1.3×) | 67.5 (↑) |
| | | | 500 | 5,621 | 0.69 | 14.3 | 1.71 (1.5×) | 70.0 (↑) |
| | LLaVA-Llama3-8B | N/A | 2,278 | 7,647 | 0.94 | 16.2 | 1.16 | 70.9 |
| | + GAZEVLM | ≥1,200 | 1,200 | 6,717 | 0.82 | 15.9 | 1.36 (1.2×) | 68.9 |
| | | | 1,000 | 6,705 | 0.82 | 15.8 | 1.38 (1.2×) | 69.0 |
| | | | 500 | 6,339 | 0.78 | 15.7 | 1.49 (1.3×) | 67.1 |

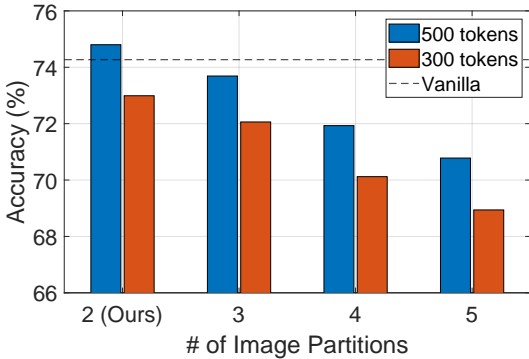

Figure 10: Effect of the number of image partitions.

significant efficiency, i.e., lower throughput, higher latency, and higher memory usage, with only minor accuracy reductions. Our GAZEVLM identifies a middle balance point under this trade-off.

**Difference number of image partitions in GAZEVLM-PRE.** We also investigate whether two partitions of global-view and local-view images are sufficient to understand the scene. We applied our token dropping into different number of partitions, where the size of each partition is decreased by a factor of $\alpha$, e.g., the selected sizes for three partitions will be a factor of 1 (our global-view image), $\frac{1}{\alpha}$ (ours local-view image), and $\frac{1}{\alpha^i}$ for $i$-th partitions. Similar to the long-context LLM effect in Fig. 3, using more partitions does not necessarily improve accuracy, as depicted in Fig. 10. The accuracy adversely decreases as the number of partitions increases, suggesting that adding more partitions introduces dilution and distraction and thus leads to accuracy drop. This implies that our double-view architecture prevents this, thereby less harming the accuracy, while saving resources.

**GAZEVLM-POST on a single view image.** To verify whether a single image (either the global-view image-only or the local-view image-only) is enough, we report the accuracy and average latency using GAZEVLM-POST on each single image (note that without our GAZEVLM-POST, the token budget cannot be ensured). As shown in Table 9, applying GAZEVLM-POST on global-view or local-view images consistently reduces latency by reducing the number of tokens, compared to *Vanilla*. The latency improvement from enough budget to the 300-token budget may appear marginal, since each single image contains only 576 tokens even under the enough budget case.

Table 9: Accuracy (Latency) of GAZEVLM-POST on global-view and local-view images.

|  | Enough Budget | 500 tokens | 300 tokens |
|---|---|---|---|
| Vanilla | 74.3 (0.77) | N/A | N/A |
| Global-view only + GAZEVLM-POST | 71.4 (0.54) | 70.1 (0.52) | 69.4 (0.48) |
| Local-view only + GAZEVLM-POST | 72.2 (0.52) | 72.6 (0.51) | 72.3 (0.47) |
| GAZEVLM (Ours) | 76.4 (0.62) | 74.8 (0.49) | 73.0 (0.45) |

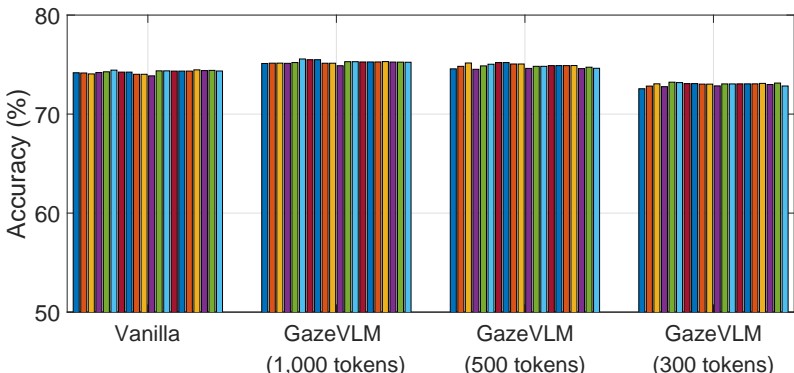

Figure 11: Leave-one-user-out testing against user bias.

Table 10: Performance on different GPUs with 100 iterations on the same sample.

| GPU | Approach | # Visual Token | Avg Latency (s) | Memory (GB) | Power (W) | Throughput (token/s) |
|---|---|---|---|---|---|---|
| L40S | Vanilla | 2,144 | 1.31 | 14.7 | 216.69 | 0.77 |
|  | GAZEVLM | 1,000 | 1.20 | 14.4 | 195.18 | 0.84 |
|  |  | 500 | 0.57 | 14.3 | 187.77 | 1.80 |
|  |  | 300 | 0.55 | 14.2 | 179.54 | 1.85 |
| P40 | Vanilla | 2,144 | 21.81 | 14.7 | 82.51 | 0.05 |
|  | GAZEVLM | 1,000 | 9.43 | 14.4 | 90.62 | 0.10 |
|  |  | 500 | 3.63 | 14.3 | 98.58 | 0.28 |
|  |  | 300 | 2.62 | 14.2 | 105.09 | 0.38 |
| A2 | Vanilla | | Out-of-Memory | | | |
|  | GAZEVLM | 1,000 | 4.33 | 14.4 | 60.19 | 0.23 |
|  |  | 500 | 1.90 | 14.3 | 60.14 | 0.53 |
|  |  | 300 | 1.62 | 14.2 | 60.26 | 0.62 |

However, as described in Fig. 2, throughput (which correlates with latency) increases significantly until the token budget is reduced to 500 and then saturates. Therefore, the improvement from 576 to 300 tokens is not significant, whereas latency compared to *Vanilla* using 2,296 tokens is substantial: latency is highly reduced by 38% and 39% for global-view-only and local-view-only, respectively, comparable to the 42% saving in GAZEVLM. Although we can still provide efficiency via GAZEVLM-POST, the accuracy with single-view images cannot achieve the accuracy as good as GAZEVLM. The global-view-only case fails to focus on user-attentive regions, while the local-view-only case is sensitive, vulnerable to gaze data with errors or low-saliency.

**User bias.** To evaluate whether GAZEVLM over-relies on specific users' gaze patterns, we conduct leave-one-user-out testing by excluding each participant from the 20 participants in the AiR-D dataset. As depicted in Fig. 11, the accuracy across all held-out user cases shows robustness across various token budgets and cross-user variabilities with an accuracy gap of less than 0.5%, which demonstrates cross-user generalizability that we do not suffer from gaze or user bias.

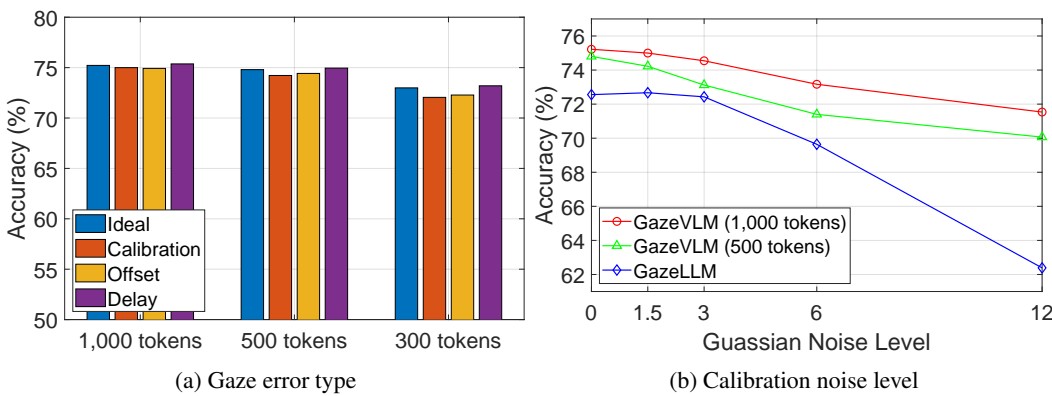

(a) Gaze error type

(b) Calibration noise level

Figure 12: Effect of inaccurate gaze data by calibration, offset, and delay errors.

**Diverse GPU resources.** Although A100-80GB is a high-end resource, the absolute latency or throughput values do not fundamentally affect our claims; the relative efficiency improvement from token reduction remains consistent across different hardware platforms. To validate this, we tested GAZEVLM on three other GPUs with different specifications: (i) NVIDIA L40S (modern high-end); (ii) Tesla P40 (mid-range, older Pascal architecture with low computing power); and (iii) NVIDIA A2 (entry-level GPU designed for edge inference (NVIDIA, 2025)). For each GPu, we measured the latency, memory usage, throughput, and additionally power consumption by averaging 100 iterations on the same sample. First of all, it should be noted that memory usage or accuracy remains the same across all platforms, as the model parameters, token budget, and selected tokens do not change. As shown in Table 10, we consistently make latency, memory, power consumption, and throughput efficient across all GPUs. In particular, on the A2 GPU, which is designed for edge scenarios, the *Vanilla* without any token selection is not runnable due to memory constraints; however, GAZEVLM with efficient token selection enables the inference. This highlights the importance of our token selection for resource-constrained edge deployment. The power consumption on the A2 GPU shows limited improvement across budgets, as the A2 GPU is optimized for energy-efficient edge-side inference workloads. Furthermore, *Vanilla* on the P40 GPU, which takes around 20 seconds (also reported in prior work such as HiRED (Arif et al., 2025)), can be reduced up to 2–4 seconds for each inference on GAZEVLM while not sacrificing accuracy. These results demonstrate that GAZEVLM provides robust efficiency across GPU platforms and shows the feasibility on practical edge scenarios.

### A.3.2 GAZE ANALYSIS

**Error in gaze data.** There can be a natural error in eye tracking data in three types: (i) calibration; (ii) offset; and (iii) delay. First, the calibration error is when the gaze point deviates from the actual point. As this can happen in any direction, we add the Gaussian Noise $\mathcal{N}(\mu, 0)$ to both $x$ and $y$ coordinates. As reported in (Holmqvist et al., 2012), we use $\mu = 1.5$ degrees as a default setting, which is the median error of the eye tracking signal by Project Aria Glasses (Engel et al., 2023; Wilson et al., 2025). Second, the offset error represents a systematic bias in a specific direction, e.g., always shifted left. To mimic this error, we add $\mathcal{N}(1.5, 0)$ only to the $x$ coordinate. Lastly, the delay error comes from the latency between when the eye actually moves and when the system computes. Typical delays range from 45ms to 81ms (Stein et al., 2021), and thus we use the gaze point from 81ms earlier. The Ideal is the original gaze data without any noise injection. As shown in Fig. 12a, these minor errors, such as 1.5 degrees (corresponding to approximately 23 pixels) or 81ms delay, are very marginal, and thus the overall accuracy is almost not impacted. With a tighter 300-token budget, accuracy drops slightly but still less than 1%, since we have limited tokens for the gaze-centered area, and those 23 pixels slightly more affect the performance.

We also investigate the effect of the noise level by varying the mean $\mu$ from 0.0 to 12.0 degrees. As illustrated in Fig. 12b, while cropping-based GazeLLM is highly degraded with the larger error, we provide much more robust inference across token budgets. This is because we hold both focused

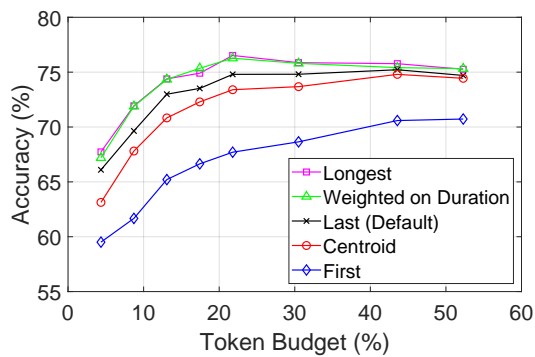

Figure 13: Performance on various gaze extraction strategies.

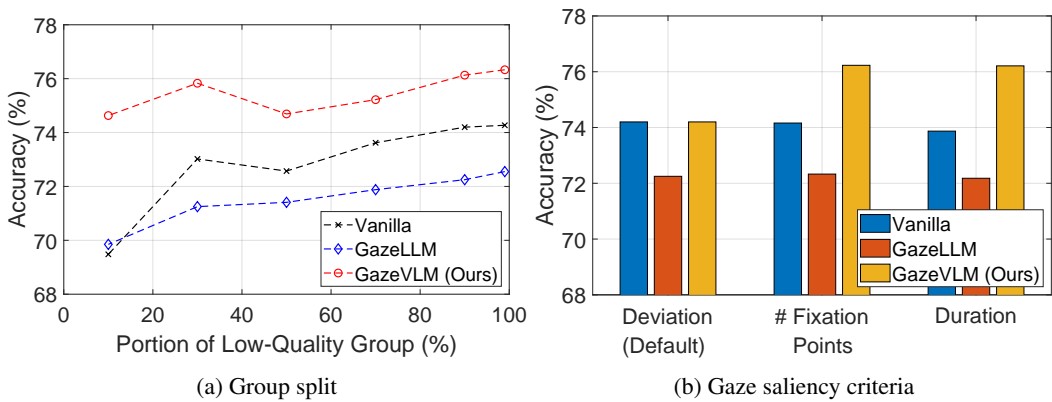

| (a) Group split | (b) Gaze saliency criteria |
|---|---|

Figure 14: Accuracy on the low-saliency group with different group splits and gaze saliency criteria.

and global understanding, while cropping-based methods discard all areas outside the gaze region, which makes those approaches highly affected by the gaze errors.

**Gaze extraction strategies.** We investigate several gaze extraction strategies that capture spatial or temporal information, including (i) the last fixation (our choice); (ii) the first fixation; (iii) the centroid by averaging all fixation points; (iv) the fixation point with the longest duration; and (v) the weighted average of fixation points based on the duration. We adopt the last fixation point for simplicity and its empirically good performance among spatial-based strategies. As depicted in Fig. 13, in the relaxed budget, there is not much difference except for using the first fixation method. Overall, selecting the longest fixation point or the duration-weighted point consistently shows the most effective. As the budget becomes tight, the accuracy gap increases; however, the gap between the last fixation point that we used with those two approaches is only 1.4% at most under a very tight 300-token budget, and therefore, we use the last fixation points for simplicity and computational efficiency.

**Gaze saliency group.** We mainly used the deviation of fixation points from the centroid as our gaze saliency measure and defined the top 10% least-deviated points as a high-saliency group. This is to highlight the limitation of cropping-based approaches, i.e., low accuracy under the low-saliency group, which motivates our design in Sec. 3. To further validate our motivation, we first evaluate the performance by varying the ratio between high-saliency and low-saliency groups. As illustrated in Fig. 14a, the cropping-based methods like GazeLLM consistently show the accuracy degradation in low-saliency groups across different ratios, while GAZEVLM can provide robust accuracy even under low-saliency gaze data. We also investigate the grouping saliency using different criteria: (i) the number of fixation points; and (ii) the average duration, where the more fixation points and shorter durations are considered as less focused and thus classified as the low-saliency group. As depicted in Fig. 14b, cropping-based GazeLLM consistently exhibits the accuracy drop compared

Table 11: Question types of AiR-D dataset. The description is extracted from the original GQA paper (Hudson & Manning, 2019).

| Type | | Description |
|---|---|---|
| Global | Object (`obj`) | For existence questions (e.g., "Is there an apple in the picture?") |
| | Global (`global`) | About overall properties of the scene such as weather or place (e.g., "How is the weather in the image?") |
| Local | Attribute (`attr`) | Consider the properties or position of an object (e.g., "What color is the apple?") |
| | Category (`cat`) | Related to object identification within some class (e.g., "What kind of fruit is on the table?") |
| | Relation (`rel`) | For questions asking about the subject or object of a described relation (e.g., "What is the girl wearing?") |

Table 12: Accuracy on different question types (from Table 11). The arrows indicate improvements compared to *Vanilla*.

| Approach | Global Type | | | Local Type | | |
|---|---|---|---|---|---|---|
| | All | Low-Saliency | High-Saliency | All | Low-Saliency | High-Saliency |
| Vanilla | 88.3 | 87.5 | 93.9 | 71.3 | 71.5 | 69.6 |
| Cropping (GazeLLM) | 87.1 | 86.9 | 88.6 | 69.5 | 69.3 | 71.5 (↑) |
| GAZEVLM | 91.5 (↑) | 90.7 (↑) | 97.2 (↑) | 73.2 (↑) | 73.2 (↑) | 73.0 (↑) |

to the *Vanilla* of the original architecture in the low-saliency gaze data across all grouping criteria, which supports our robust, balanced design between the focused yet the global context.

**Question types.** We investigate the accuracy under different question types. The QA set in the AiR-D dataset originates from the GQA dataset (Hudson & Manning, 2019). Following the definitions of question types in GQA, we categorize them into two types as described in Table 11: (i) Global Type: questions requiring holistic understanding, including 1,876 samples; and (ii) Local Type: questions on a certain object, including 8,943 samples. As shown in Table 12, the cropping-based GazeLLM shows the improvement only for the Local Type questions under the high-saliency gaze data. It implies that gaze-guided cropping is effective when the questions are about a certain object, and the gaze data can indicate the object of interest. However, gaze-aware cropping becomes harmful if questions require the global scene understanding or when the gaze data is inaccurate, as the cropped image may exclude critical visual contents. In contrast, GAZEVLM consistently improves accuracy across both question types and gaze data saliency, which highlights our robustness.

**Analysis on question and image types.** We analyze the generated answers across different question types and images across four cases in Table 13: (i) question with well-aligned gaze; (ii) question on a small image area with well-aligned gaze; (iii) question on a small image area with less-aligned gaze; and (iv) question requiring global image understanding. In the first case (in the second column of Table 13), the gaze is concentrated and well-aligned, and the target object is visually large enough. Therefore, all approaches of Vanilla, GazeLLM (cropping), and GAZEVLM perform well. For the second case (in the third column), the target object is small, and thus the Vanilla (without any gaze integration) fails to capture the user attentive area, while other gaze-guided methods can successfully focus on the target area along with the well-aligned gaze. For the third and fourth cases (in the fourth and the fifth columns, respectively), GazeLLM, which removes the surrounding information and has only a local-view image, fails to capture the object of interest when gaze is less accurate (in the third case) or when the question needs global context (in the fourth case). In contrast, GAZEVLM holds both focused and global understanding, which makes the inference robust across various images and question types.

Table 13: Example of the image with a heatmap for different question or image types.

| | | | | |
|---|---|---|---|---|
| Global-View Image (Original) |  |  |  |  |
| Local-View Image (Cropped) |  |  |  |  |
| Gaze Heatmap |  |  |  |  |
| Overlaid Image |  |  |  |  |
| Question | Are there carts near the pond | What type of vegetable is to the right of the knife | What is the basket filed with | Are there both glasses and ties in the picture |
| Answer (Ground-Truth) | yes | peppers | tennis balls | yes |
| Answer (Vanilla) | **Yes**, there is a cart near the pond ... | (✗) ... it's not clear which specific vegetable is | ... filled with **tennis balls** | **Yes**, there are both glasses and ties ... |
| Answer (GazeLLM) | **Yes**, there is a cart near the pond ... | ... with several green bell **peppers** ... | (✗) ... shows a tennis court with a player... | (✗) No, there are no glasses ... |
| Answer (Ours) | **Yes**, there is a cart near the pond ... | ... appears to be a green bell **pepper** | ... filled with **tennis balls** | **Yes**, there is a person ... wearing glasses and a tie |

## A.4 DISCUSSION

**Gaze data.** As we rely on gaze data to identify regions of interest, we are naturally affected by the quality (e.g., calibration, offset, or delay errors) or the saliency (e.g., consistency or user interest) of eye-tracking data and how to extract the gaze data from scanpath, i.e., the sequential set of gaze points. Although GAZEVLM showed the robust inference even under low-quality gaze, there is still

a natural gap between low-saliency and high-saliency groups. This suggests that adapting the design to gaze saliency could further enhance robustness.

**Parameter tuning.** As shown in Tables 12 and 13, results are affected by the question types, e.g., whether the question needs global understanding or not. Although we have shown robust yet efficient inference by balancing the global context and the gaze-focused understanding, we can further investigate principled designs for parameter tuning. For example, we can dynamically set the parameters based on the gaze saliency to give more budget to gaze tokens when having better saliency.

**Limited datasets.** The only datasets that integrate real eye-tracking records into VQA are AiR-D and VQA-MHUG, which we used. However, they are constructed by displaying images on screens or XR devices and then collecting the real-world eye gaze data, not real-world QA sets in smart glasses scenarios. Therefore, we plan to collect the real-world VQA dataset with synced eye gaze.

**Edge platforms.** As explained in Table 10, we have shown the efficiency of GAZEVLM under A2 GPU, which is generally used for edge use cases (NVIDIA, 2025). In future work, we will deploy our framework on real-world eyewear devices.

