# OpenReview forum: "GazeVLM: Gaze-Guided Vision-Language Models for Efficient and Robust Inference"
_ICLR.cc/2026/Conference — Submitted to ICLR 2026_

### Official Review · Reviewer_48ay · 2025-10-27

**Soundness:** 3
**Presentation:** 2
**Contribution:** 2
**Rating:** 2
**Confidence:** 5

**Summary:**

"GAZEVLM: GAZE-GUIDED VISION-LANGUAGE MODELS FOR EFFICIENT AND ROBUST INFERENCE" proposes a novel vision-language model (VLM) framework,  which incorporates gaze information to improve the model's inference efficiency and robustness on resource-constrained devices. Traditional VLMs typically require processing a large number of visual tokens, resulting in high latency and low throughput, making real-time interaction difficult. GAZEVLM addresses this issue through a two-stage strategy:

GAZEVLM-PRE (pre-processing stage): Extracts local and global view images based on the user's gaze point, balancing details with overall information and enhancing robustness to low-quality eye movement data.

GAZEVLM-POST (post-processing stage): After image encoding, visual tokens are prioritized based on gaze point, retaining key tokens in the gaze region while uniformly sampling tokens from other regions to meet the token budget constraint.

**Strengths:**

A gaze-guided token selection framework is proposed for efficient VLM inference. It selects visual tokens directly based on the user's gaze area, improving efficiency and preserving user intent.

A gaze-aware image preprocessing mechanism was designed, combining local and global views to improve the model's robustness to variations in eye movement data quality and avoid the loss of critical visual information.

A token selection strategy combining gaze and global information is proposed, which reduces the number of tokens while maintaining global understanding capabilities. It is suitable for deployment on edge devices such as smart glasses.

**Weaknesses:**

The paper lacks innovation in its integration of gaze and token selection. It recommends strengthening the method's uniqueness by introducing a gaze-attention fusion mechanism and comparing their complementarity.

The experiment only roughly categorizes gaze quality. Realistic noise such as offset, delay, and calibration error, is needed to validate robustness.

The paper ignores gaze bias and cross-user generalization. Leave-one-user-out testing and heatmap visualization should be conducted to diagnose whether the model over-relies on specific gaze patterns.

A static 0.5 token allocation ratio cannot adapt to content and question variations.

The model relies solely on an offline static dataset. The model needs to be deployed on edge devices such as smart glasses, and conduct user experience studies to demonstrate its efficiency and credibility in real-world scenarios.

**Questions:**

Please provide statistics on the overlap between gaze and the ground-truth region, as well as the accuracy drop when the IoU between gaze and ground-truth regions is less than 0.3.

The experiment only categorizes high and low quality images based on gaze offset and does not inject system noise.

Currently, a fixed β of 0.5 is used to assign gaze/surrounding tokens, but the need for local-global information varies significantly between different problems.

All experiments were conducted offline on static images, lacking end-to-end latency, energy consumption, and user subjective evaluations on real smart glasses.

The paper only reports a 1.9× throughput improvement for 500 tokens, but does not disclose the contribution of the LLM stage to total latency.

How can a circular gaze mask be aligned with the square patch boundary?

---

> ### Author Response · Authors · 2025-11-21
> **Response to Reviewer 48ay**
>
> We sincerely thank the reviewer for your thorough and constructive feedback. We carefully addressed all points and confirmed that our claims remain consistent. Based on the suggestions, we have updated as follows:
>
> * (W1) Method's uniqueness: added clarification on novelty and comparison with attention-based methods
> * (W2, Q2) Gaze error and gaze quality: added new results in Figs. 12 and 14
> * (W3) User bias: added leave-one-user-out testing in Fig. 11 and heatmaps in Table 13
> * (W4, Q3) Question variations: added results on different question types in Tables 11-13
> * (W5-1, Q4) Limited datasets: started to collect new data using smart glasses and will include as many results as possible before the rebuttal ends
> * (W5-2, Q4) Edge devices: added evaluations on edge-level GPU and demonstrated the feasibility
> * (Q5) Latency decomposition: full latency decomposition across token budgets will be added before rebuttal ends
> * (Q6) Gaze masking: clarified and added illustrative examples in Fig. 5
>
> ## ***W1: Method's Uniqueness***
>
> Our novelty lies in the design: GazeVLM is the first to integrate eye gaze for efficient and robust VLM inference and to introduce gaze-guided token selection.
>
> While gaze-attention fusion is an interesting idea, our experimental results in Table 2 show that attention-based HiRED (61.4%) performs worse than our global-token-only sampling (74.6% at $\beta=0$ in Fig. 9(b)) and simple Spatial (72.2%), which samples tokens uniformly regardless of attention scores or gaze. This is because the attention score tends to capture visually important or large objects rather than user intent. Based on this, a gaze-attention fusion (e.g., gaze-HiRED) is likely to perform worse than our gaze-global fusion.
>
> ## ***W2, Q2: Gaze Error and Gaze Quality***
>
> Based on the reviewer's suggestion, we additionally evaluated other criteria such as the number of fixation points and fixation duration in Fig. 14(b) (see Appendix A.3.2) and demonstrated that GazeVLM maintains robust accuracy even under low-saliency gaze.
>
> We have also newly added the performance under noisy gaze data (including offset, delay, and calibration errors) in Fig. 12(a) and different noise levels in Fig. 12(b) (see Appendix A.3.2) and verified that GazeVLM consistently provides robust accuracy in various token budgets.
>
> ## ***W3: User Bias***
>
> To evaluate cross-user generalization, we newly conducted the suggested leave-one-user-out testing on the AiR-D dataset with 20 participants in Fig. 11 and the heatmap visualization in Table 13 (see Appendices A.3.1 and A.3.2, respectively). The results show that excluding any single user does not affect the accuracy (less than 0.5% across different token budgets), which implies that GazeVLM does not over-rely on specific users' gaze patterns.
>
> ## ***W4, Q3: Question Variations***
>
> The optimal local-global token allocation may vary depending on the question or content variations. To investigate this, we newly evaluated performance across Global Type (e.g., "Is there an apple in the picture") and Local Type (e.g., "What color is the apple?") questions in Tables 11-13 (see Appendix A.3.2). Notably, GazeVLM with $\beta=0.5$ consistently improves accuracy across question types, demonstrating robustness to variations.
>
> We further investigated the best $\beta$ for each type as follows:
>
> | | Global Type | Local Type |
> |:---:|:---:|:---:|
> | 1,000 tokens | 0.5 | 0.7 |
> | 500 tokens | 0.3 | 0.5 |
>
> Based on the results, we observed that Global Type questions benefit from more surrounding tokens with smaller $\beta$, while Local Type questions prefer more gaze-focused tokens with larger $\beta$. Overall, effective values still lie within $0.3 \leq \beta \leq 0.7$, as we claimed in Fig. 9(b) (see Appendix A.3.1), which supports our default 0.5 as a robust compromise.
>
> ## ***W5-1, Q4: Limited Datasets***
>
> At the time of submission, the two VQA datasets used were the only publicly available ones with real eye-tracking data. We agree that evaluating on streaming, real-time scenarios is important, and *we are currently starting to collect new datasets using Meta Aria Glasses*. Additionally, we plan to incorporate QA sets from Project Aria (e.g., Aria Everyday Activities) or Ego4D, which provide egocentric data with eye gaze collected on smart glasses, to better align our evaluation. We will include results on these datasets as much as possible before the rebuttal ends.

---

> > ### Author Response · Authors · 2025-11-21
> > **Response to Reviewer 48ay (2)**
> >
> > ## ***W5-2, Q4: Edge Devices***
> >
> > While our experiments primarily used A100-80GB, the relative efficiency improvements from token selection are consistent across hardware platforms. We further tested GazeVLM on three additional GPUs: NVIDIA L40S (high-end); Tesla P40 (mid-range but with old NVIDIA Pascal architecture and low computing power); and A2 (entry-level used for edge use cases [1]). As shown in Table 10, GazeVLm consistently improves latency, memory, throughput, and power efficiency. Notably, Vanilla is not runnable on the edge-setting A2 GPU due to memory constraints; however, GazeVLM enables inference, which highlights our feasibility for edge deployment.
> >
> > Due to OS and system constraints, full deployment on smart glasses and user studies will be addressed as our next piece of work.
> >
> > [1] NVIDIA, "NVIDIA A2 Tensor Core GPU – Entry-level GPU that brings NVIDIA AI to any server," Available at: https://www.nvidia.com/en-us/data-center/products/a2/ (Accessed Nov 20, 2025).
> >
> > ## ***Q1: Overlap between Gaze and the Ground-Truth Region***
> >
> > We thank the reviewer for the suggestion, and it will be very interesting to analyze the results. However, ground-truth regions are not available in the current datasets, so we are not able to compute IoU between gaze and target objects. Instead, we evaluated robustness under varying gaze error in Figs. 12 and 14 (see Appendix A.3.2), which simulate less-aligned gaze. These results show that GazeVLM maintains high accuracy even when gaze is less reliable, which implies that we may maintain the accuracy even when IoU is small. We try to sample several examples and approximate target regions using object detection or light manual annotation and report qualitative results if time permits.
> >
> > ## ***Q5: Latency Decomposition***
> >
> > In a typical VLM pipeline, the LLM stage dominates the total latency: the ViT encoder takes 136.9ms, while the LLM takes 392.8ms on a lightweight 2B VLM [2]. The number of visual tokens will affect the prefilling stage (i.e., to compute Key-Value (KV) cache). We will prepare a full latency decomposition across token budgets will be included before the rebuttal ends.
> >
> > [2] Huang, Jin, et al. "LiteVLM: A Low-Latency Vision-Language Model Inference Pipeline for Resource-Constrained Environments." arXiv preprint arXiv:2506.07416 (2025).
> >
> > ## ***Q6: Gaze Masking***
> >
> > Since token selection is based on the Euclidean distance from the gaze point, the boundary may be slightly cropped at the boundary. We have added illustrative examples in Fig. 5 (see Appendix A.1).

---

### Official Review · Reviewer_9pQx · 2025-10-31

**Soundness:** 3
**Presentation:** 3
**Contribution:** 2
**Rating:** 4
**Confidence:** 4

**Summary:**

This paper proposes GAZEVLM, a framework to improve the inference efficiency and robustness of Vision-Language Models (VLMs), particularly for real-time applications on devices like smart glasses. The core problem it addresses is the high latency and memory usage caused by processing an excessive number of visual tokens.
The proposed solution, GAZEVLM, uses eye gaze as a proxy for user intent and introduces a two-phase mechanism:
1. GAZEVLM-PRE: A gaze-aware preprocessing step that, instead of just cropping, generates two views: a "global-view" image (full scene) and a "local-view" image (gaze-centered crop). This is designed to maintain global context for robustness, inspired by foveated rendering.
2. GAZEVLM-POST: A gaze-guided token selection step after encoding. It selects a subset of tokens from both views to meet a specific token budget $T_b$. This selection prioritizes "gaze tokens" (near the gaze point) while also sampling "surrounding tokens" to retain context.
The authors evaluate GAZEVLM on two VQA datasets with real eye-tracking data (AiR-D and VQA-MHUG). The results show that GAZEVLM can achieve up to 1.9x higher throughput and 37% lower latency (with a 500-token budget, ~22% of vanilla) while slightly improving accuracy compared to the full-token baseline.

**Strengths:**

1. The paper proposes a method for efficient inference in VLMs while maintaining robustness, enabling their deployment on resource-constrained platforms.
2. The paper presents a novel two-phase (PRE/POST) architectural design. This design is motivated by key insights from prior work: 1) existing gaze-guided methods (e.g., pure cropping) are vulnerable to low-quality gaze data, and 2) existing efficient VLM methods (e.g., attention-based token dropping) may not align with user intent.
3. The method achieves impressive efficiency. The experiments demonstrate that with only 22% of the original tokens, the architecture can achieve higher accuracy, 1.9x higher throughput, and maintain robustness.

**Weaknesses:**

1. The novelty of the individual components is limited. GAZEVLM-PRE is essentially a combination of "center cropping + global scaling," and GAZEVLM-POST combines "Euclidean distance-based selection + uniform sampling." These techniques, in isolation, are not new.

2. The performance of the baselines seems questionable. The HiRED baseline performs exceptionally poorly (61.4% accuracy), and the paper's justification (misalignment with user intent) is a strong claim that may not be fully supported. An unfair implementation is a possible alternative explanation. More importantly, the GAZEVLM-PRE ablation (1,200 tokens) already outperforms the Vanilla baseline (2,295 tokens) in both accuracy (76.4% vs. 74.3%) and throughput (2.02 vs. 1.42). This implies that the 2-view foveated partitioning (PRE) might be the main source of improvement, rather than the POST token selection. The paper attributes the gains to the combined PRE+POST framework, but the individual contributions are not clearly disentangled.

3. There is a significant mismatch between the motivation and the experimental setup. The paper motivates the work with real-time, streaming applications (XR, VR, autonomous driving), but the evaluation is conducted on static VQA datasets. These datasets are limited, and the methodology of using only the "last gaze point" is an oversimplification that seems arbitrary and is not well-justified.

**Questions:**

1. Please explain the discrepancy between the 2,880 tokens for the "Vanilla" baseline mentioned in the text (Footnote 1, Appendix A.2) and the 2,295 tokens used in all experimental tables (Table 1, 3, 4)? Which number is correct, and how are the efficiency gains calculated?
2. The definition of "gaze deviation" used for the robustness study (Fig 4) is unclear. The split ("top 10% low deviation" vs. "the rest") feels arbitrary. How was "gaze deviation" calculated? What was the gaze deviating from? Would you consider a more systematic robustness evaluation by simulating noise on the gaze coordinates?
3. The GAZEVLM-PRE ablation (Table 3) outperforms the Vanilla baseline in both accuracy and efficiency. Does this imply that the LLaVA-Next 5-view partitioning is simply a poor design and that your 2-view foveated partitioning is the main source of improvement, rather than the GAZEVLM-POST token selection? Is the comparison between a 5-partition and 2-partition architecture a fair baseline?
4. The data processing uses only the last gaze point, which is a significant simplification. What is the justification for this? Why not use the point with the longest fixation duration, or the centroid of the entire scanpath?
5. The HiRED baseline performs very poorly. Could you please clarify its implementation?
6. Can the practical significance of the hyperparameters $\alpha$ and $\beta$ be explained beyond a simple parameter sweep? How do these two parameters interact, and do they determine how the PRE and POST modules collaborate?

---

> ### Author Response · Authors · 2025-11-21
> **Response to Reviewer 9pQx**
>
> We truly thank the reviewer for the constructive feedback. We have updated our manuscript as follows:
> * (W1) Novelty of the individual components: clarified the contribution and explored alternative designs to justify our design choice
> * (W2-1, Q5) HiRED baseline: clarified HiRED performance
> * (W2-2, Q3) GazeVLM-Pre vs. GazeVLM-Post vs. Vanilla: clarified the differences
> * (W3-2, Q4) Gaze extraction strategies: added Fig. 13 to show the impact of different gaze extraction strategies
> * (Q1) The number of tokens in Vanilla and efficiency gain calculation: explained how to compute the number of tokens and efficiency gain
> * (Q2-1) Gaze deviation: clarified the definition for the gaze deviation
> * (Q2-2) Gaze split: added Fig. 14(a) with different grou splits
> * (Q2-3) Gaze error: added Figs. 12(a) and 12(b) for noisy gaze data
> * (Q6) Hyperparameter tuning: compared with adaptive parameter tuning strategies and analyzed what affects the parameter tuning
>
> ## ***W1: Novelty of the Individual Components***
> We also explored alternative designs, such as gaze-attention fusion (see W2-1) or adaptive $\alpha$ and \beta$ based on gaze saliency (see Q6), but either performed worse or similar to our design. These results confirm that our design choices are robust, effective, and simple, while further principled, adaptive, and optimized strategies can be a promising direction for future work.
>
> However, our novelty lies in the design: GazeVLM is the first to integrate eye gaze for efficient and robust VLM inference and to introduce gaze-guided token selection.
>
> ## ***W2-1, Q5: HiRED Baseline***
> We used the open source of the HiRED GitHub page and did not modify the method except for the input data parts.
>
> As further evidence from the HiRED paper, HiRED maintains reasonable accuracy under a limited token budget on visually simple datasets, such as VQAv2 (80.3%$\rightarrow$77.5%) and the slightly more complex TextVQA (64.8%$\rightarrow$61.4%), where attention scores can easily capture visually important objects (most VQAv2 images contain only one or a few objects, as shown in Table 5). However, *HiRED severely fails in complex benchmarks*, such as DocVQA (73.4%$\rightarrow$60.8%) or ChartQA (54.8%$\rightarrow$42.0%). For example, in the first example of Table 4, the visually dominant object is a boy in front, but the question asks about a small knife behind him. In such cases, attention-based approaches fail to capture user-intended regions, making them perform poorly.
>
> ## ***W2-2, Q3: GazeVLM-Pre vs. GazeVLM-Post vs. Vanilla***
> The reviewer's observation is correct. The 5-partition Vanilla baseline is static and gaze-agnostic, which may fragment objects of interest across multiple partitions and fail cross-partition attentions. In contrast, our gaze-aware 2-partitioning centers on user-attended regions, enhancing full coverage of objects, which improves accuracy. Therefore, the improvement of GazeVLM-Pre is from gaze-guided, context-aware partitioning rather than GazeVLM-Post. GazeVLM-Pre also provides better throughput than Vanilla since GazeVLM-Pre starts from two partitions rather than five. However, as shown in Fig. 3, GazeVLM-Pre alone (green dot) cannot satisfy a token budget and thus cannot ensure efficiency.
>
> To tackle the budget problem, GazeVLM-Post of token selection is designed primarily for efficiency, as discussed in Sec. 2.1. Moreover, GazeVLM-Post without GazeVLM-Pre of the 2-view foveated partitioning (blue line) shows lower accuracy than the full GazeVLM (red line). The combined Pre+Post design is therefore necessary to achieve both robust accuracy and efficient inference.
>
> ## ***W3-1: Limited Datasets***
> At the time of submission, the two VQA datasets were the only publicly available ones with real eye-tracking data. We agree that evaluating on streaming, real-time scenarios is important, and we are currently starting to collect new datasets using Meta Aria Glasses. Additionally, we plan to incorporate QA sets from Project Aria (e.g., Aria Everyday Activities) or Ego4D, which provide egocentric data with eye gaze collected on smart glasses, to better align our evaluation. We will include results on these datasets as much as possible before the rebuttal ends.
>
> ## ***W3-2, Q4: Gaze Extraction Strategies***
> We have newly added Fig. 13 (see Appendix A.3.2) to investigate the impact of different gaze extraction strategies, including (i) first fixation; (ii) centroid; (ii) last fixation; (iv) longest fixation; and (v) duration-weighted fixation. As shown in Fig. 13, the last fixation showed the best accuracy among spatial-only strategies. Although temporal strategies slightly outperform our last fixation (up to 1.4% at a tight token budget), temporal strategies require storing all gaze points with duration, and thus we use the last fixation points for simplicity and computational efficiency. We are happy to update the results using the longest or duration-weighted approaches if encouraged.

---

> > ### Author Response · Authors · 2025-11-21
> > **Response to Reviewer 9pQx (2)**
> >
> > ## ***Q1: The Number of Tokens in Vanilla and Efficiency Gain Calculation***
> > We apologize for the confusion. The Vanilla baseline can generate up to 2m880 tokens (576 tokens$\times$5 partitions), but the actual number of visual tokens injected to the LLM varies depending on the input image resolution. For non-square images, LLaVA adds some dummy padding tokens to fit a fixed size e.g., 336$\times$336 pixels) and then be removed after the image encoding stage. Therefore, each image has different ratios between the width and the height, and thus the number of actual visual tokens injected to LLM is different. In short, 2,880 tokens is the maximum possible (in cases of square-shaped images), while 2,295 tokens are the actual number used on average.
> >
> > Regarding the efficiency gain computation, with an example with a 500-token budget from RQ 1 on Table 3, the throughput gain of 1.9$\times$ is computed as:
> > $$\frac{Ours}{Vanilla}=\frac{2.70}{1.42} \approx 1.9,$$
> > while the reduced latency of 36% is calculated as follows:
> > $$\frac{Ours-Vanilla}{Vanilla}\times 100=\frac{0.49-0.77}{0.77} \times 100 \approx -36.36\%.$$
> > The memory usage is a simple gap between the Vanilla and our GazeVLM.
> >
> > ## ***Q2-1: Gaze Deviation***
> > We apologize for the missing description. The gaze deviation is defined as the standard deviation of gaze points from the centroid.
> >
> > ## ***Q2-2: Gaze Split***
> > The split between the top 10% low-deviation was chosen based on the human visual system: the macular region is 18 degrees, corresponding to 46 pixels, which matches the average deviation in the top 10% group. While we initially used 10% based on this insight, this choice is not strict. We have newly added Fig. 14(a) with different splits (see Appendix A.3.2), which consistently supports our motivation for balanced design of focused and global understanding.
> >
> > ## ***Q2-3: Gaze Error***
> >
> > We have also newly added the performance under noisy gaze data (including offset, delay, and calibration errors) in Fig. 12(a) and different noise levels in Fig. 12(b) (see Appendix A.3.2) and verified that GazeVLM consistently provides robust accuracy in various token budgets.
> >
> > ## ***Q6: Hyperparmeter Tuning***
> > To address the reviewer's question on the practical significance of the hyperparameters and how they determine the collaboration between PRE and POST modules, we further conducted a principled parameter tuning based on gaze saliency (i.e., deviation from centroid) as follows:
> >
> > $$\alpha_i = \frac{max_{\forall i}(deviation) - deviation_i}{max_{\forall i}(deviation)}, \beta_i = 1 - \frac{max_{\forall i}(deviation) - deviation_i}{max_{\forall i}(deviation)},$$
> >
> > where the smaller $\alpha$ and the larger $\beta$ is more preferred for focused understanding. As the results show below, principled $\beta$ performs comparably to our 0.5 setting, while $\alpha$ and $\beta$ perform best at Original (0.5). This confirms that our empirical choice of $\alpha=\beta=0.5$ provides a robust accuracy by balancing focused and global understanding.
> >
> > | | Original (0.5) | Principled $\alpha$ | Principled $\beta$ | Principled $\alpha$ and $\beta$ |
> > |:---:|:---:|:---:|:---:|:---:|
> > | 1,000 tokens | 75.2 (best) | 73.8 | 75.2 (best) | 74.0 |
> > | 500 tokens | 74.8 (best) | 72.9 | 74.8 (best) | 72.8 |
> > | 300 tokens | 73.0 (best) | 70.8 | 72.3 | 70.4 |
> >
> > Furthermore, the performance can vary depending on the question or image types. To investigate this, we newly evaluated performance across Global Type (e.g., "Is there an apple in the picture") and Local Type (e.g., "What color is the apple?") questions in Tables 11-13 (see Appendix A.3.2). Notably, GazeVLM with $\beta=0.5$ consistently improves accuracy across question types, demonstrating robustness to variations. We also provide the best $\beta$ for each type as follows:
> >
> > | | Global Type | Local Type |
> > |:---:|:---:|:---:|
> > | 1,000 tokens | 0.5 | 0.7 |
> > | 500 tokens | 0.3 | 0.5 |
> >
> > Based on the results, we observed that Global Type questions benefit from more surrounding tokens with smaller $\beta$, while Local Type questions prefer more gaze-focused tokens with larger $\beta$. Overall, effective values still lie within $0.3 \leq \beta \leq 0.7$, as we claimed in Fig. 9(b) (see Appendix A.3.1), which supports our default 0.5 as a robust compromise.
> >
> > Overall, these results confirm that our empirical choice of 0.5 provides robustness, while principled and adpative strategies can be a promising direction for future work.

---

> > > ### Comment · Reviewer_9pQx · 2025-11-28
> > > **Following up**
> > >
> > > Thank you for the detailed response and the additional experiments, which have resolved most of my concerns, I would consider to raise my score if the two remaining questions could be resolved.
> > >
> > > 1.Baseline Selection:
> > > Why was HiRED selected as the sole baseline for efficient VLMs? Since your related work mentions other approaches (e.g., ZipVL, MiniCache), could you clarify the rationale for this specific choice?
> > >
> > > 2.Comparison Logic:
> > > You emphasize that GAZEVLM-PRE and POST must work together to ensure both accuracy and budget control. What is the specific rationale for comparing them individually against baselines in the evaluation? Is the intent to demonstrate that neither component is sufficient on its own, or to prove that each component independently outperforms its direct counterparts (e.g., Pre > Cropping)?

---

> > > > ### Author Response · Authors · 2025-11-28
> > > > **Follow-up Response to Reviewer 9pQx**
> > > >
> > > > We thank the reviewer for raising this important point. Although the reviewer could not change the recommendation due to the website issue, we would still like to respond to the clarification.
> > > >
> > > > **Baseline selection.** We would like to clarify the rationale for baselines. First, HiRED has shown that it outperforms LLaVA-PruMerge by a large gap, and thus we compared with HiRED, a SOTA in efficient VLMs with early token dropping, i.e., dropping tokens before LLM. Second, ZipVL or FastV is not an early dropping method: it already has to load all tokens onto memory. For efficient VLMs, only dropping before LLM can satisfy the resource constraints. Additionally, MiniCache is more for efficient cache compression rather than sparse computation (i.e., token dropping). To clarify this, we have added further explanation in Sec. 2.1.
> > > >
> > > > Most approaches including HiRED are designed for general visual importance rather than user-specific context. To the best of our knowledge, there is no work of efficient VLM using the gaze as the user context; all gaze-integrated VLMs are not designed for efficiency and do not consider any token dropping techniques but only focus on gaze-centered cropping. Therefore, we select HiRED as the most preferred baseline for efficient VLM. However, we will include more recent baselines on our best if needed.
> > > >
> > > > **Comparison logic.** We aim to clarify both the individual contributions of GazeVLM-Pre and GazeVLM-post and the necessity of using them together.
> > > >
> > > > First, we show that GazeVLM-Pre alone already outperforms other approaches in terms of accuracy. However, preprocessing alone cannot satisfy the target budget; thus, GazeVLM-Post is needed for efficient inference. Overall, these comparisons are to show that (i) each module independently improves over its direct baselines, and (ii) both modules are necessary to achieve both accuracy and efficiency.

---

### Official Review · Reviewer_qxk5 · 2025-10-31

**Soundness:** 3
**Presentation:** 2
**Contribution:** 2
**Rating:** 4
**Confidence:** 3

**Summary:**

This paper introduces GazeVLM, a novel framework that leverages eye gaze data to improve the efficiency and robustness of Vision-Language Model (VLM) inference. The approach consists of two key components: (1) GazeVLM-PRE: a gaze-aware preprocessing mechanism that extracts both local-view (gaze-focused) and global-view images before encoding, and (2) GazeVLM-POST: a gaze-guided token selection method after encoding that prioritizes tokens around the gazing area while maintaining global context. The authors demonstrate that their approach achieves higher throughput and lower latency while using only 22% of tokens compared to vanilla architectures, with minimal or even improved accuracy on two VQA datasets with real eye-tracking data.

**Strengths:**

1. The paper addresses a relevant problem for deploying VLMs on resource-constrained smart glasses and XR devices, with clear motivation for using eye gaze as a user intent signal.
2. The combination of preprocessing (for robustness) and postprocessing (for budget control) is intuitive and shows practical benefits.
3. Unlike existing cropping-based approaches, the paper explicitly considers gaze quality variations and demonstrates maintained performance under low-quality gaze data.

**Weaknesses:**

1. Lack of Theoretical Foundation and Principled Design. The paper is primarily engineering-driven with limited theoretical justification for its design choices. Critical parameters such as α = 0.5 (gaze region ratio) and β = 0.5 (gaze token ratio) are selected purely through empirical grid search without principled reasoning. The even split between local/global views (T_v = ½T_a) and the choice of circular regions for gaze tokens lack any theoretical or optimization-based justification. More fundamentally, the paper provides no information-theoretic analysis of why gaze-guided selection should preserve task-relevant information, no formal characterization of the trade-off between local detail and global context, and no principled framework for when gaze guidance helps versus hurts performance. The token selection strategy based on simple Euclidean distance and uniform sampling is overly simplistic, as it ignores semantic relationships between tokens, doesn't consider token importance beyond spatial proximity, and doesn't explore learning-based or attention-weighted selection strategies.

2. Severely Limited Experimental Diversity and Scope. The experimental validation relies on only 2 datasets (AiR-D and VQA-MHUG), both collected in controlled settings rather than real-world scenarios, both relatively small (10K and 8K samples), and both limited to VQA tasks only. There is no evaluation on other critical vision-language tasks such as image captioning, visual grounding, visual reasoning, or video understanding, all of which are essential for smart glasses applications. The model coverage is equally narrow, testing only LLaVA variants (Mistral-7B, Llama3-8B) without exploring other VLM architectures or larger models where efficiency gains would be more impactful. The paper also lacks evaluation on scenarios where gaze might be ambiguous (multiple similar objects), dynamic scenes, multi-turn dialogues, or tasks requiring global reasoning (counting, spatial relationships). This limited scope raises serious concerns about the generalizability of the findings and whether the method works beyond the specific controlled settings tested.

3. Overly Simplistic and Unjustified Gaze Data Utilization. The paper's use of gaze data discards rich information that could improve performance. Using only the "last gaze point" is never justified and ignores valuable temporal patterns—scanpath sequences reveal cognitive processes, fixation durations indicate importance levels, and first versus later fixations have different semantic meanings. The binary quality split (top 10% versus rest) is arbitrary with no justification, missing opportunities for continuous modeling of gaze uncertainty or adaptive strategies based on confidence scores. The paper doesn't leverage gaze prediction uncertainty information that real eye trackers provide, doesn't employ probabilistic modeling of gaze locations, and doesn't handle gaze prediction errors in a principled way. Furthermore, there's no ablation study comparing different gaze extraction strategies (first fixation, longest fixation, centroid, duration-weighted average), making it impossible to know if the chosen approach is optimal or even reasonable.

4. Incomplete Baseline Comparisons and Missing State-of-the-Art Methods. The paper fails to compare against several recent and relevant efficient VLM methods, including LLaVA-PruMerge (which is mentioned in related work but never compared), FastV, VisionZip, VideoLLM-online, VideoLLM-MoD, and MiniCache. The comparison with HiRED appears somewhat unfair since HiRED is designed for general visual importance rather than user-specific context, yet the paper doesn't discuss scenarios where visual importance might align with user intent or explore hybrid approaches combining both signals. There's also no comparison with learned token selection methods or reinforcement learning-based approaches, which could potentially outperform the heuristic-based selection. This incomplete comparison makes it difficult to assess whether the performance gains come from gaze integration specifically or simply from having any reasonable token selection strategy, and whether more sophisticated methods could achieve better results.

5. Insufficient Analysis, Ablations, and Methodological Rigor. The paper lacks critical analyses that would provide insights into when and why the method works. There's no failure case analysis explaining when GazeVLM fails, no breakdown by question types (spatial vs. semantic), no analysis of how performance varies with image complexity, and no reporting of computational overhead for preprocessing. The ablation studies are limited—there's no exploration of different token selection strategies (circular vs. elliptical vs. attention-weighted regions), no comparison of gaze extraction methods, and no study on the optimal number of gaze points to use. Methodologically, the paper only reports accuracy without other important metrics (F1, precision-recall for specific object types), lacks human evaluation of response quality, provides no confidence intervals or significance tests for the reported improvements, and doesn't specify important details like the "deviation" metric for gaze quality assessment or random seeds for reproducibility.

**Questions:**

1. Can you provide information-theoretic or optimization-theoretic justification for your design choices? Why should gaze-guided selection preserve task-relevant information?
2. Why not learn α and β from data? Why not make them adaptive based on gaze quality, image complexity, or question type?
3. Have you compared different gaze extraction strategies (first fixation, longest fixation, centroid, weighted average by duration)? Why is "last gaze point" optimal?
4. How does your method perform on tasks requiring global reasoning (counting, spatial relationships)?
5. Why not compare with recent efficient VLM methods like LLaVA-PruMerge, FastV, or VisionZip? How does your method compare to learned token selection?
6. In what scenarios does gaze guidance hurt performance? When should we prefer gaze-agnostic methods?

---

> ### Author Response · Authors · 2025-11-21
> **Response to Reviewer qxk5**
>
> We appreciate the reviewer's comments to point out our important aspects. Based on suggestions, we have newly added as follows:
> * (W1-1, Q1, Q2, Q4, Q6) Principled design: added the comparison to principled design and Tables 11-13
> * (W1-2) Token selection strategy and token importance: newly evaluated a different token selection strategy
> * (W2-1) Limited datasets: started to collect new data using smart glasses and will include before rebuttal ends
> * (W3-1) Gaze extraction strategies: added Fig. 13 of different gaze extraction strategies
> * (W3-2) Gaze quality split: newly added Figs. 14(a)-(b)
> * (W3-3) Gaze error: newly included Figs. 12(a)-(b)
> * (W4, Q5) Baselines: clarified why we select HiRED as a baseline
> * (W5-2) Computational overhead for preprocessing: reported the preprocessing latency below
>
> ## ***W1-1, Q1, Q2, Q4, Q6: Principled Design***
> To explore principled parameter design, we used adaptive parameters based on gaze deviation (from centroid) as follows:
>
> $$\alpha_i = \frac{max_{\forall i}(deviation) - deviation_i}{max_{\forall i}(deviation)}, \beta_i = 1 - \frac{max_{\forall i}(deviation) - deviation_i}{max_{\forall i}(deviation)},$$
>
> where the smaller $\alpha$ and the larger $\beta$ is more preferred for focused understanding. As the results show below, principled $\beta$ performs comparably to our 0.5 setting, while $\alpha$ and $\beta$ perform best at Original (0.5). This confirms that our empirical choice of $\alpha=\beta=0.5$ provides a robust accuracy by balancing focused and global understanding.
>
> | | Original (0.5) | Principled $\alpha$ | Principled $\beta$ | Principled $\alpha$ and $\beta$ |
> |:---:|:---:|:---:|:---:|:---:|
> | 1,000 tokens | 75.2 (best) | 73.8 | 75.2 (best) | 74.0 |
> | 500 tokens | 74.8 (best) | 72.9 | 74.8 (best) | 72.8 |
> | 300 tokens | 73.0 (best) | 70.8 | 72.3 | 70.4 |
>
> Furthermore, the performance can vary depending on the question or image types. To investigate this, we newly evaluated performance across Global Type (e.g., "Is there an apple in the picture") requiring global reasoning and Local Type (e.g., "What color is the apple?") questions in Tables 11-13 (see Appendix A.3.2). Notably, GazeVLM with $\beta=0.5$ consistently improves accuracy across question types, demonstrating robustness to variations. We also provide the best $\beta$ for each type as follows:
>
> | | Global Type | Local Type |
> |:---:|:---:|:---:|
> | 1,000 tokens | 0.5 | 0.7 |
> | 500 tokens | 0.3 | 0.5 |
>
> Based on the results, we observed that Global Type questions benefit from more surrounding tokens with smaller $\beta$, while Local Type questions prefer more gaze-focused tokens with larger $\beta$. Overall, effective values still lie within $0.3 \leq \beta \leq 0.7$, as we claimed in Fig. 9(b) (see Appendix A.3.1), which supports our default 0.5 as a robust compromise.
>
> Overall, these results confirm that our empirical choice of 0.5 provides robustness, while principled and adpative strategies can be a promising direction for future work.
>
> ## ***W1-2: Token Selection Strategy and Token Importance***
> Our Euclidean-based token selection strategy is inspired by fixed foveated rendering (FFR), which is commonly used in XR/VR applications. Square-shaped regions are also used in eye-tracked foveated rendering (ETFR), so we additionally evaluated performance under square shapes as follows:
>
> | | 1,000 tokens | 500 tokens | 300 tokens | 100 tokens |
> |:---:|:---:|:---:|:---:|:---:|
> | Circle (Our Choice) | 75.2 | 74.8 | 73.0 | 66.1 |
> | Square | 75.3 (+0.1) | 75.0 (+0.2) | 72.4 (-0.6) | 65.9 (-0.2) |
>
> As shown above, the square shape is slightly better at 500-1,000 tokens, while the circular shape is preferred for tighter budgets at 100-300 tokens. Overall, the differences are marginal, and the important aspect is to include the gaze-near area. Thus, our Euclidean distance-based selection is simple but effective in gaze-integrated methods.
>
> We do not use attention-based methods. As evidenced from the HiRED paper, HiRED maintains reasonable accuracy under a limited token budget on visually simple datasets, such as VQAv2 (80.3%$\rightarrow$77.5%) and the slightly more complex TextVQA (64.8%$\rightarrow$61.4%), where attention scores can easily capture visually important objects (most VQAv2 images contain only one or a few objects, as shown in Table 5). However, *HiRED severely fails in complex benchmarks*, such as DocVQA (73.4%$\rightarrow$60.8%) or ChartQA (54.8%$\rightarrow$42.0%). For example, in the first example of Table 4, the visually dominant object is a boy in front, but the question asks about a small knife behind him. In such cases, attention-based approaches fail to capture user-intended regions, making them perform poorly. We also avoid learning-based selection, which requires dataset-dependent training or fine-tuning.

---

> ### Author Response · Authors · 2025-11-21
> **Response to Reviewer qxk5 (2)**
>
> ## ***W2-1: Limited Datasets***
> We are currently starting to collect new datasets using Meta Aria Glasses. Additionally, we plan to incorporate QA sets from Project Aria or Ego4D. We will include results on these datasets as much as possible before the rebuttal ends.
>
> ## ***W2-2: Limited Tasks and Models***
> As shown in prior work, we believe that our approach can be applied to various vision-language tasks and models. We will try our best to include the results of other tasks or models.
>
> ## ***W2-3: Limited Evaluation Scenarios***
> Due to the time constraints, we have newly included the evaluation under different types of whether requiring global reasoning in Tables 11-13. Please see the details in W1-1.
>
> ## ***W3-1: Gaze Extraction Strategies***
> We have newly added Fig. 13 (see Appendix A.3.2) to investigate the impact of all suggested gaze extraction strategies. In Fig. 13, the last fixation showed the best accuracy among spatial-only strategies. We have not considered temporal strategies; we sincerely appreciate the reviewer's insightful comments. Based on the suggestion, we observed that temporal strategies slightly outperform our last fixation (up to 1.4% improvement at a tight token budget); however, temporal strategies require storing all gaze points with duration, and thus we keep the last fixation points for simplicity and computational efficiency. We are happy to update the results using the longest or duration-weighted approaches if encouraged.
>
> ## ***W3-2: Gaze Quality Split***
> The split between the top 10% low-deviation was chosen based on the human visual system: the macular region is 18 degrees, corresponding to 46 pixels, which matches the average deviation in the top 10% group. However, this choice is not strict. We have newly added Fig. 14(a) with different splits (see Appendix A.3.2), which consistently supports our motivation for balanced design of focused and global understanding.
>
> ## ***W3-3: Gaze Error***
> We have also added results under noisy gaze data (including offset, delay, and calibration errors) in Fig. 12(a) or noise levels in Fig. 12(b) (see Appendix A.3.2) and verified that GazeVLM is consistently robust across various token budgets.
>
> ## ***W4, Q5: Baselines***
> We would like to clarify on baselines. First, HiRED has shown that it outperforms LLaVA-PruMerge by a large gap, and thus we compared with HiRED, a SOTA in efficient VLMs with early token dropping, i.e., dropping tokens before LLM. Second, FastV is not an early dropping method: it already has to load all tokens onto memory. For efficient VLMs, only dropping before LLM can satisfy the memory constraints. To clarify this, we have added further explanation in Sec. 2.1. Third, VideoLLM-online and VideoLLM-MoD have a different task from ours. Fourth, MiniCache is more for efficient cache compression rather than sparse computation (i.e., token dropping).
>
> As the reviewer exactly pointed out, HiRED is designed for general visual importance rather than user-specific context. To the best of our knowledge, there is no work of efficient VLM using the gaze as the user context; all gaze-integrated VLMs are not designed for efficiency and do not consider any token dropping techniques but only focus on gaze-centered cropping. Therefore, we select HiRED as the most preferred baseline for efficient VLM. However, we will include more recent baselines on our best.
>
> ## ***W5-1: Analysis on GazeVLM Failures and Question Types***
> Due to the time constraints, we conducted the experiments under different types of question types and illustrative examples in Tables 11-13. Please see the details in W1-1.
>
> ## ***W5-2: Computational Overhead for Preprocessing***
> As the preprocessing is based on the simple cropping centered with gaze point, it is quickly done using OpenCV libraries. We averaged the preprocessing latency over all instances, and it takes only 6.9 milliseconds on average, which is relatively negligible.
>
> ## ***W5-3: Others***
> Our deviation metric for gaze quality is the deviation from the centroid. Regarding the reproducibility, we built our codebase using ```transformers``` library from HuggingFace, and we use the default settings. All the random seeds are fixed; we have confirmed that all accuracy values remain the same across different platforms.
>
> Please check the results across different token selection strategies in W1-2, the comparison of gaze extraction methods in W3-1, and the optimal number of gaze points (tuned by parameter $\beta$) in W1-1. Lastly, we will try our best to include other metrics (e.g., F1 or precision-recall as the reviewer suggested) before the rebuttal ends.

---

### Official Review · Reviewer_BFCe · 2025-11-01

**Soundness:** 3
**Presentation:** 4
**Contribution:** 4
**Rating:** 6
**Confidence:** 5

**Summary:**

The authors present a novel training-free efficient VLM inference framework called GazeVLM. GazeVLM utilizes gaze information for a visual stimulus to ascertain important tokens that must be utilized by the VLM to respond to a textual prompt. This allows the VLM to operate under strict token budget constraints, thereby gaining throughput and reducing memory footprint, while not sacrificing performance. Through experiments and analysis, the authors claim that GazeVLM is able to outperform previous methods in terms of efficacy and efficiency.

**Strengths:**

(1) Great motivation: I find the core intuition of using gaze to make VLMs more token-efficient in terms of the vision modality relevant and useful. Making VLMs more context-aware, i.e., understand user’s visual attention, while participating in conversation with the user is also a task worth studying.

(2) Adequate model design: I find the way the authors have analyzed the limitations of existing efficient VLMs and proposed a solution to mitigate these limitations worthy of appreciation. The training-free approach is both straightforward and effective.

(3) Thorough experimental analysis: The authors have performed a very thorough and comprehensive set of experiments and ablations in both the main text and the appendix.

**Weaknesses:**

(1) Method might be proved redundant by better vision-language alignment: Even though the solution is novel and well-motivated, as VLMs get more sophisticated (as shown in Appendix A.3), e.g., learn to choose visual tokens that are more relevant to the prompt, explicitly using gaze might lose its efficacy. In an early study, it was revealed that between 70-95% of  fixated objects are described in the corresponding language descriptions. In that case, learning better vision-language alignment might be sufficient to deduce useful visual tokens.

(2) Performance gains are not significant: Simple baselines like Pooling (2 X 2) and Dotted Map are not significantly outperformed by GazeVLM in efficiency/efficacy metrics. I wonder for other simpler tasks, like object localization, if these simple baselines may fare against GazeVLM. From Figure 4, Circled Map does better than GazeVLM for high quality gaze. However, we expect eye trackers to get more sophisticated, and expect gaze data to be better quality as days go by.

**Questions:**

(1) What happens if you don’t provide the global-view tokens and force GazeVLM to rely only on local-view image?

(2) Any strong reason for choosing not to fine-tune/retrain a VLM like Voila-A but using the core intuition of the paper?

---

> ### Author Response · Authors · 2025-11-21
> **Response to Reviewer BFCe**
>
> We appreciate the reviewer recognizing our insightful motivation and thorough experimental analysis. We carefully addressed all comments and confirmed that all our claims remain the same. Based on the suggestions, we have updated as follows:
> * (W1) Vision-language alignment: explained the vague description and the efficiency problems
> * (W2) Comparison to simple baselines: explained the benefits of GazeVLM and clarified factors that can affect the gaze
> * (Q1) Local-only results: added Table 9
> * (Q2) Plug-and-play: clarified the benefit of plug-and-play in terms of easy adoption and superior accuracy of foundation models
>
> ## ***W1: Vision-Language Alignment***
> While VLMs may learn to select visual tokens relevant to a prompt, the problem lies in the vague and less-specified prompt itself. In natural language, users often do not describe objects in full detail, e.g., the red cup with the bear on it on the table; instead, they use vague descriptions, e.g., that cup (Yan et al., 2024). Therefore, vision-language alignment fails to reliably identify the object under such vague descriptions or cluttered views. In contrast, gaze directly indicates what the user is attending.
>
> Furthermore, efficiency is another critical problem. The correlation between the prompt and the image can be computed within the LLM [1] by injecting all tokens into the LLM first, and it requires loading all tokens onto memory. For efficient VLMs, dropping before LLM is necessary to meet resource constraints.
>
> [1] Ye, Hanrong, et al. "MM-Ego: Towards Building Egocentric Multimodal LLMs for Video QA." arXiv preprint arXiv:2410.07177 (2024).
>
> ## ***W2: Comparison to Simple Baselines***
> While simple baselines such as Pooling (2$\times$2) and Dotted Map may achieve comparable efficiency, they show the large accuracy degradation with the gap of up to 5.3% in Table 3. (Please note that the improvement by introducing a dynamic partitioning in LLaVA was 1.4% and 1.8% for GQA and VQA-v2 datasets, respectively, which are the original QA sets of AiR-D and VQA-MHUG. Thus, 5.3% degradation is substantial.) Compared to Dotted Map, the problem lies in the efficiency rather than the accuracy, where the throughputs of Dotted Map and our GazeVLM (with even better accuracy) are 1.42 tokens/s and 2.70 tokens/s, respectively. Circled Map may slightly outperform GazeVLM under the high-saliency group, but the difference is <1% while using 1.9$\times$ more tokens, 24% longer latency, and 26% poorer throughput. Furthermore, our GazeVLM works better in overall accuracy (ours of 76.4% vs. Circled Map of 74.7%).
>
> As the reviewer mentioned, the error of the gaze data itself can be resolved as days go by. However, the low-saliency group (we redefine to low-saliency group rather than the low-quality group) is not only affected by the eye-tracking error but also by the user's eye movement behaviors, such as saccades, pursuits, or fixations [2]. Some users tend to saccade more rather than fixate at a single point, while others do not move their eyes quickly. These different user eye patterns can be another reason.
>
> Also, the performance can vary depending on question types: Global Type (e.g., "Is there an apple in the picture") and Local Type (e.g., "What color is the apple?") questions in Tables 11-13 (see Appendix A.3.2). Notably, GazeVLM consistently improves accuracy across question types, which demonstrates robustness to diverse variation factors.
>
> [2] Oyama, Akane, et al. "Novel method for rapid assessment of cognitive impairment using high-performance eye-tracking technology." Scientific reports 9.1 (2019): 12932.
>
> ## ***Q1: Local-Only Results***
> The local-view alone is only a single cropped, low-resolution image, which cannot achieve a good accuracy due to the low-resolution (which motivates to design of a dynamic partitioning in recent works, as described in Sec. 2.1) and the sensitivity under low-saliency gaze data. To verify this, we have newly added Table 9 (see Appendix A.3.1) to apply GazeVLM-Post on the local-view image-only case, and we have confirmed that local-view only with GazeVLM-Post (with the accuracy of 72.6% at most) works worse than GazeVLM (with the accuracy of 76.4% at most), which demonstrates the necessity of having tokens from both global-view and local-view.
>
> ## ***Q2: Plug-and-Play***
> We propose a plug-and-play without any model (re-)training or changes; we make computation efficient via token selection. There are two main reasons. First, it enables easy adoption on existing VLMs with minimal integration effort. Second, importantly, we can maintain superior accuracy. The reason that we use foundation models trained on vast data is due to their powerful adaptability to many downstream tasks or benchmarks. However, if we (re-)train or fine-tune the model, the existing foundation models become skewed or biased to the fine-tuned datasets, which make the model less generalizable and loses their power of having superior accuracy.

---

> > ### Comment · Reviewer_BFCe · 2025-11-23
> > **Following up**
> >
> > Thanks to the authors for the clarification. However, some concerns still remain, and I would like to follow up on the authors' responses:
> >
> > 1. With respect to W1: Like I said in my review, in an early study [1] (apologies for missing the reference in my review), it was revealed that between 70-95% of fixated objects are described in the corresponding language descriptions, and conversely 70-90% of objects in descriptions are fixated. This shows a great overlap between gaze and language. Following up on this, even though users may issue an ambiguous command like "that cup", how often does that happen in the datasets studied in the paper?
> >
> > 2. With respect to W2: The authors seem to have missed by query about how well their method might fare on simpler tasks like localization, when compared to the simple baselines. Can the authors comment on this query (just to clarify, I am not asking for additional experiments)?
> >
> > The authors also appear to attribute low performance to "different user eye patterns". However, this is expected for a complex task like VQA. How do the authors plan on mitigating this issue?
> >
> > Finally, can the authors elaborate how the poor performance for low-saliency group and aberrant user behavior affect usage in real life scenarios?
> >
> > [1] Yun, Kiwon, et al. "Studying relationships between human gaze, description, and computer vision." Proceedings of the IEEE Conference on Computer Vision and Pattern Recognition. 2013.

---

> > > ### Author Response · Authors · 2025-11-25
> > > **Follow-up Response to Reviewer BFCe**
> > >
> > > We appreciate the reviewer's quick follow-up and clarification. We have carefully addressed your questions, and we would be happy to respond to any further comments.
> > >
> > > ## ***Gaze-Language Alignment***
> > >
> > > > With respect to W1: Like I said in my review, in an early study [1] (apologies for missing the reference in my review), it was revealed that between 70-95% of fixated objects are described in the corresponding language descriptions, and conversely 70-90% of objects in descriptions are fixated. This shows a great overlap between gaze and language.
> > >
> > > We appreciate the clarification and the reference. As the reviewer pointed out, the early study reports that 72-95% of fixated objects align with the description [1]. However, **this result is for image description/captioning/annotation tasks, where tasks themselves are explicitly asked to describe objects**. This is fundamentally different from VQA, where users often provide short, ambiguous questions rather than full descriptions. Moreover, in description tasks, the annotator should mention objects, while in VQA, especially in real-world smart glasses scenarios, questions rarely include explicit object names; instead, they will be vague, e.g., "What is that building?," and those objects may not even be named at all, e.g., "What is this?" [2], [3]. Therefore, the high gaze-description overlap in [1] does not directly imply the gaze-question overlap for VQAs.
> > >
> > > ## ***Ambiguity***
> > >
> > > > Following up on this, even though users may issue an ambiguous command like "that cup", how often does that happen in the datasets studied in the paper?
> > >
> > > Ambiguity is a well-known core and common feature of natural language, as well as in VQA datasets [4], [5]. Even when an object is mentioned in the question, images often include multiple candidate objects (e.g., "What is next to the mirror?" but having multiple mirrors in the image). Quantitatively, **in VQAv2 (the QA sets used in the AiR-D studied in our paper), those multiple instances of the same category account for 85%** [5].
> > >
> > > ## ***GazeVLM on Simple Tasks***
> > >
> > > > With respect to W2: The authors seem to have missed by query about how well their method might fare on simpler tasks like localization, when compared to the simple baselines. Can the authors comment on this query (just to clarify, I am not asking for additional experiments)?
> > >
> > > We appreciate the clarification and apologize for missing the reviewer's question regarding simpler tasks. For simpler, localized tasks like object localization, the accuracy gap between GazeVLM and simple baselines is expected to be smaller, as these tasks rely mainly on local spatial information. However, **even in our Local Type questions (targeting a specific local object), GazeVLM still outperforms the baseline (73.2% for GazeVLM vs. 69.5% for GazeLLM in Table 12)**, although the gap is smaller than that of Global Type questions (3.7% in Local Type vs. 4.3% in Global Type). This implies that ours can provide accuracy gains even for relatively simpler localized tasks.
> > >
> > > ## ***Poor Performance***
> > >
> > > > The authors also appear to attribute low performance to "different user eye patterns". However, this is expected for a complex task like VQA. How do the authors plan on mitigating this issue? Finally, can the authors elaborate on how the poor performance for low-saliency group and aberrant user behavior affect usage in real life scenarios?
> > >
> > > **We would like to clarify that low performance in terms of both accuracy and efficiency is the problem in baselines, not in ours**. Simple baselines of cropping- or highlighting-based approaches can mistakenly discard or miss the actual target region. In contrast, our approach already mitigates this issue by maintaining global coverage with focused understanding; thus, our GazeVLM provides good accuracy under low-saliency group (+1.9% in our GazeVLM vs. -2.6% for the baseline of GazeLLM in Fig. 4) and also in a simple, localized question task (+1.9% in our GazeVLM vs. -1.8% for the baseline of GazeLLM in Table 12). Importantly, efficiency is still the problem for existing gaze-aware baselines; they do not ensure the token budget and thus cannot provide any efficiency. In short, GazeVLM is already designed for both efficient and robust VLM inference.
> > >
> > > [2] Piantadosi, Steven T., et al. "The communicative function of ambiguity in language." Cognition 122.3 (2012): 280-291.
> > >
> > > [3] Pezzelle, Sandro. "Dealing with Semantic Underspecification in Multimodal NLP." Proceedings of the 61st Annual Meeting of the Association for Computational Linguistics (Volume 1: Long Papers). 2023.
> > >
> > > [4] Stengel-Eskin, Elias, et al. "Why did the chicken cross the road? rephrasing and analyzing ambiguous questions in vqa." Proceedings of the 61st Annual Meeting of the Association for Computational Linguistics (Volume 1: Long Papers). 2023.
> > >
> > > [5] Chen, Chongyan, et al. "Acknowledging Focus Ambiguity in Visual Questions." Proceedings of the IEEE/CVF International Conference on Computer Vision. 2025.

---

> > > > ### Comment · Reviewer_BFCe · 2025-11-27
> > > > **Following up again**
> > > >
> > > > Thanks for the clarification - that resolves only some of my concerns, while the rest remain:
> > > >
> > > > (1) Ambiguity: The authors claimed "Quantitatively, in VQAv2 (the QA sets used in the AiR-D studied in our paper), those multiple instances of the same category account for 85% [5].". However, the paper referenced says that "... with 85% (i.e., 3792/4440) belonging to visual questions with a single answer grounding  and 15% (i.e., 648/4440) belonging to visual questions with multiple answer groundings." Can the authors clarify?
> > > >
> > > > (2) Vision-Language Alignment: The authors have consistently claimed that "For efficient VLMs, dropping before LLM is necessary to meet resource constraints.". Using gaze as additional guidance sounds useful in some scenarios, I wonder if the overhead of additional eye tracking hardware and software will offset the efficiency gains in the VLM. Can the authors comment on this?
> > > >
> > > > (3) Poor Performance: I want to clarify that I was referring to the second paragraph in the authors' initial rebuttal response, where they talk about "These different user eye patterns can be another reason". Can the authors comment on this variability within users and how that can be resolved? Also wanted to clarify that I was referring to the low *gains* in performance in my initial review, not the low performance.

---

> > > > > ### Author Response · Authors · 2025-11-28
> > > > > **Second Follow-Up Response to Reviewer BFCe**
> > > > >
> > > > > Thank you again for the follow-up and clarification. We would be happy to further address any additional questions you may have.
> > > > >
> > > > > **(1) Ambiguity**: Thank you for the correction. These are the extracted sentences from [5]:
> > > > >
> > > > > > 85% (i.e., 3792/4440) belonging to visual questions with a single answer grounding and 15% (i.e., 648/4440) belonging to visual questions with multiple answer groundings
> > > > >
> > > > > > Multiple instances of the same category account for 61.5% overall, with 84.6% (i.e., 77) in VQAv2
> > > > >
> > > > > In short, the aforementioned 84.6% (among 91 instances only from VQAv2) is the percentage of multiple instances of the same category *among ambiguous samples*, while **overall ambiguous questions account for 15%** (among 4,440 instances from VizWiz-VQA and VQAv2 datasets), as the reviewer pointed out. We apologize for the confusion.
> > > > >
> > > > > **(2) Vision-Language Alignment**: Thank you for pointing out the practical aspect. As the reviewer pointed out, there can exist the overhead of additional eye tracking hardware and software. However, **the overhead is not a big concern relative to VLM inference**. Actually, this delay (defined as "the time it takes for the eye tracker to register a change in eye position and deliver this information to the experimental software" [6]) of recent eye trackers is typically 15-52 ms [6], whereas VLM inference operates on seconds level timescale, e.g., our GazeVLM with 500 tokens (which does not harm accuracy) reduces the latency from 0.77$\rightarrow$0.49 seconds on a powerful A100 GPU to 21.81$\rightarrow$3.63 seconds on a low-computing P40 GPU (see Tables 2 and 10). Therefore, even under such overhead, our efficiency remains clear.
> > > > >
> > > > > **(3) Poor Performance**: We thank the reviewer for the clarification. Regarding the variability in user eye patterns, our intention was to note that such variability can lead to more-deviated gaze points. As shown in Fig. 4, GazeVLM remains robust across both more-deviated and less-deviated groups, achieving accuracy gains of +1.9% and +3.4%, respectively. As the reviewer correctly pointed out, under those variabilities, the accuracy gains are actually low; however, while higher accuracy is desirable, our main goal is to provide efficient inference without sacrificing accuracy, where our GazeVLM achieves efficiency and robustness across variability groups.
> > > > >
> > > > > **This variability can be further resolved through adaptive allocation strategies.** For example, allocating more budget to global tokens under more-deviated gaze patterns can achieve 75.3% for more-deviated groups under a 1,000-token budget, compared to 74.2% with our default setting. **Additionally, temporal smoothing or different gaze aggregation/extraction strategies can be further adopted.** For example, as shown in Fig. 13 (see Appendix A.3.2), selecting the fixation point with the longest duration can reduce the effect of fluctuated gaze patterns and achieve a slightly better accuracy (up to 1.4% in a very tight budget). Although both gaps are not significant ($\approx$ 1%), it suggests that further optimized balancing or aggregation techniques can be promising directions (while our current balanced design already provides simple yet effective robustness).
> > > > >
> > > > > [6] Stein, Niklas, et al. "A comparison of eye tracking latencies among several commercial head-mounted displays." i-Perception 12.1 (2021): 2041669520983338.

---

### Official Review · Reviewer_zVVV · 2025-11-03

**Soundness:** 2
**Presentation:** 2
**Contribution:** 2
**Rating:** 4
**Confidence:** 3

**Summary:**

The paper integrates gaze into VLM inference with a two-stage approach, GAZEVLM-Pre and GAZEVLM-Post. In GAZEVLM-Pre, the authors create two views of each image: a global full-scene view and a local view cropped around the gaze point. In GAZEVLM-Post, they apply token-selection strategies to keep only a subset of vision tokens, reducing the token budget to improve efficiency and enable deployment on wearable devices. They compare their approach against several baselines on two VQA datasets with real human eye-tracking (AiR-D and VQA-MHUG).

**Strengths:**

The paper tackles a real problem in modern VLMs: high inference cost that prevents practical use in interactive AR/smart-glasses. Reducing visual tokens while preserving accuracy is a sensible way forward. The authors propose a simple, effective method that lowers memory usage and increases throughput without sacrificing accuracy.

**Weaknesses:**

The paper could be clearer. There’s repeated text between the background and introduction, and the method section needs more plain, consistent notation. The post-gaze stage is under-explained: it’s not clear how tokens are picked, what rules guide the choice, or how the token budget is enforced. It’s also not obvious why a simple gaze-centered crop wouldn’t work just as well. Both approaches need tuning (token budget/ratios vs. crop size), so a direct, well-tuned comparison to a cropping baseline would help show the real benefit.

The evaluation is limited to two lab-style datasets. Results on truly egocentric, in-the-wild data would make the claims stronger—e.g., testing on Ego4D (if suitable gaze annotations are available).

**Questions:**

1. Could you run a controlled study where you inject noise into the gaze coordinates and plot accuracy vs. noise level for GAZEVLM and the baselines?

2. In POST, you use a fixed 50/50 split between gaze-near tokens and uniformly sampled “surrounding” tokens. Is that always best? Did you try learning this allocation or adapting it by question type (e.g., “What color is the sign I’m looking at?” might want mostly gaze tokens, while “Where are we?” needs more global tokens). Also, do you have stats on how token sources vary by image/question type?

3. How do you avoid redundant/duplicate visual tokens between the global and local views? Since the global view already contains the local region, why encode both views and then select tokens from each, instead of encoding only the global image and dropping tokens far from the gaze center?

4. Could a global-only + gaze-aware token selection baseline match your accuracy/latency without the second (local) pass? If not, can you show an ablation?

5 . All experiments are on an A100-80GB with batch size 1, which is a very strong setup. For the smart-glasses story, what on-device or edge hardware are you actually targeting, and is the 300–500 token regime small enough for real-time on those devices (latency, memory, and power)? If possible, can you share end-to-end numbers on a representative edge platform?

---

> ### Author Response · Authors · 2025-11-21
> **Response to Reviewer zVVV**
>
> We appreciate the reviewer's thorough comments. We carefully addressed all comments and confirmed that all our claims remain the same. Based on the suggestions, we have updated as follows:
> * (W1-1) Clear writing: revised repeated text and notations and added Table 3, Algorithm 2, and Fig. 5 to clearly explain our token selection
> * (W1-2) Justification for poor performance of cropping: clarified the limitations of simple cropping methods
> * (W2) Limited datasets: started to collect new data using smart glasses and will include as many results as possible before the rebuttal ends
> * (Q1) Noise injection: added Fig. 12 under different noise types and levels
> * (Q2) Varying image or question types: added results on different types in Tables 11-13 and optimal $\beta$ across types
> * (Q3) Duplicate tokens between the global and local views: explained the difference between local tokens and global tokens
> * (Q4) Global-only results: added Table 9
> * (Q5) Edge settings: added evaluation on edge-level GPU and demonstrated the feasibility
>
> ## ***Weakness 1-1: Clear Writing***
> We have fully read the paper carefully and revised the repeated text. Also, we have checked notations and newly included Table 3 (see Appendix A.1).
>
> Furthermore, we have newly added Algorithm 2 and Fig. 5 (see Appendix A.1) on how to pick tokens. To enforce the token budget, we enforce stopping sampling when the budget is out, as shown in Lines 10-12 of Algorithm 2.
>
> ## ***Weakness 1-2: Justification on Poor Performance of Cropping***
> There are two problems in simple cropping methods: (i) sensitivity under gaze qualities; and (ii) failure in capturing the full object. First, they are sensitive to the gaze data quality; if the gaze point is inaccurate, cropping-only approaches lose all the information. Therefore, as shown in Table 1 in Sec. 3, cropping methods (i.e., GazeLLM and GazeGPT in the last two rows) fail in low-quality gaze data. Second, the size of the object of interest differs. Therefore, if the object to crop is large, the cropping methods fail to fully include the whole object, which makes the model hard to understand the object. Therefore, cropping-only methods cannot ensure robustness. (Additionally, cropping-only methods before image encoding cannot provide efficient inference, since the token budget can be only ensured through postprocessing, as described in Sec. 2.1.)
>
> As evidence, as the reviewer suggested, we also investigated the accuracy by varying the cropping size ($α$) as follows:
>
> | Cropping Size ($α$) | 0.9 | 0.8 | 0.7 | 0.6 | 0.5 | 0.4 | 0.3 | 0.2 | 0.1 |
> |:---:|:---:|:---:|:---:|:---:|:---:|:---:|:---:|:---:|:---:|
> | GazeVLM (Ours) | 73.8 | 74.1 | 74.9 | 75.5 | 76.4 (best) | 76.0 | 75.8 | 74.9 | 73.5 (worst) |
> | GazeLLM (Cropping) | 74.2 | 74.1 | 74.3 (best) | 73.4 | 72.6 | 70.8 | 68.3 | 63.9 | 52.0 (worst) |
>
> As shown in the table, the cropping-only method is highly sensitive to the cropping size, ranging from 52.0% to 74.3%, while ours ranges from 73.5% to 76.4%, which justifies that simple cropping methods are less robust. Also, the accuracy even with a well-tuned GazeLLM of 0.7 is 74.3%, which is still lower than 76.4% of ours (while GazeLLM even uses 2.44$\times$ more tokens).
>
> ## ***Weakness 2: Limited Datasets***
> At the time of submission, the two VQA datasets were the only publicly available ones with real eye-tracking data. We agree that evaluating on streaming, real-time scenarios is important, and we are currently starting to collect new datasets using Meta Aria Glasses. Additionally, we plan to incorporate QA sets from Project Aria (e.g., Aria Everyday Activities) or Ego4D that the reviewer suggested, which provide egocentric data with eye gaze collected on smart glasses, to better align our evaluation. We will include results on these datasets as much as possible before the rebuttal ends.
>
> ## ***Question 1: Noise Injection***
> We have newly added the accuracy under different noise levels in Fig. 12(b) (see Appendix A.3.2). While the cropping baseline of GazeLLM is significantly affected by injecting higher noise (72.6%$\rightarrow$62.4%), we verified that our GazeVLM consistently provides robust accuracy in various token budgets (75.2%$\rightarrow$71.5% at 1,000 tokens and 74.8%$\rightarrow$70.1% at 500 tokens). We additionally evaluated with different types of errors (including offset, delay, and calibration errors) in Fig. 12(a) (see Appendix A.3.2).

---

> ### Author Response · Authors · 2025-11-21
> **Response to Reviewer zVVV (2)**
>
> ## ***Question 2: Varying Image or Question Types***
> As the reviewer pointed out, the optimal split can vary depending on the question or image types. To investigate this, we newly evaluated accuracy across Global Type (e.g., "Is there an apple in the picture") and Local Type (e.g., "What color is the apple?") questions in Tables 11-13 (see Appendix A.3.2). Notably, GazeVLM with $\beta=0.5$ consistently improves accuracy across question types, demonstrating robustness to variations. We further investigated the best split ($\beta$) for each type as follows:
>
> | | Global Type | Local Type |
> |:---:|:---:|:---:|
> | 1,000 tokens | 0.5 | 0.7 |
> | 500 tokens | 0.3 | 0.5 |
>
> Based on the results, we observed that Global Type questions benefit from more surrounding tokens with smaller $\beta$, while Local Type questions prefer more gaze-focused tokens with larger $\beta$. However, overall, effective values still lie within $0.3 \leq \beta \leq 0.7$, as we claimed in Fig. 9(b) (see Appendix A.3.1), which supports our default 0.5 as a robust compromise.
>
> Furthermore, we also conducted an adaptive parameter tuning based on gaze saliency (i.e., deviation from the centroid) as follows:
> $$\alpha_i = \frac{max_{\forall i}(deviation) - deviation_i}{max_{\forall i}(deviation)}, \beta_i = 1 - \frac{max_{\forall i}(deviation) - deviation_i}{max_{\forall i}(deviation)},$$
> where the smaller $\alpha$ and the larger $\beta$ is more preferred for focused understanding. As the results show below, principled $\beta$ performs comparably to our 0.5 setting, while $\alpha$ and $\beta$ perform best at Original (0.5). This confirms that our empirical choice of $\alpha=\beta=0.5$ provides a robust accuracy by balancing focused and global understanding.
>
> | | Original (0.5) | Principled $\alpha$ | Principled $\beta$ |
> |:---:|:---:|:---:|:---:|
> | 1,000 tokens | 75.2 (best) | 73.8 | 75.2 (best) |
> | 500 tokens | 74.8 (best) | 72.9 | 74.8 (best) |
>
> Overall, these results confirm that our empirical choice of 0.5 provides robustness.
>
> ## ***Question 3: Duplicate Tokens between the Global and Local Views***
> The tokens from the global and local views are clearly different. There are two reasons.
>
> | | Global-View | Local-View |
> |:---:|:---:|:---:|
> |Pixels in Original Image | 672$\times$672 | 336$\times$336 |
> |Pixels in Resized Image | 336$\times$336 | 336$\times$336 |
> |Patch size on Resized Image | 14$\times$14 | 14$\times$14 |
> |Patch size on Original Image | 28$\times$28 | 14$\times$14 |
>
> First, each patch includes a different granularity of the image. Specifically, given the original input image of 672$\times$672 pixels (see Fig. 6 in Appendix A.2.1), the global-view image is resized into 336$\times$336 pixels from the original 672$\times$672 pixels, which becomes 24$\times$24=576 patches, where each patch can include 14$\times$14 pixels, corresponding to 28$\times$28 pixels in the original image. On the other hand, the local view image with a cropping size of 0.5 extracts (not resizes) 336$\times$336 pixels, and thus each patch can include 14$\times$14 pixels in the original image. Therefore, patches in the global and local views include different granularity.
>
> Second, more importantly, tokens are computed by the image encoding. Therefore, token embeddings from the global view capture full-image context, while token embeddings from the local view reflect the cross-attention on the cropped region.
>
> ## ***Question 4: Global-only Results***
> The global-only itself has only a single low-resolution image, which cannot have a good accuracy (which motivates to design a dynamic partitioning in recent works, as described in Sec. 2.1). To verify this, we have newly added Table 9 (see Appendix A.3.1) to apply GazeVLM-Post on the global-view image-only case. We have confirmed that the latency is consistently reduced, while the accuracy of a global-only (of 71.4% at most) is worse than ours (of 76.4% at most) using both global-view and local-view images.
>
> ## ***Question 5: Edge Settings***
> While our experiments primarily used A100-80GB, the relative efficiency improvements from token selection are consistent across hardware platforms. We further tested GazeVLM on three additional GPUs: NVIDIA L40S (high-end); Tesla P40 (mid-range but with old NVIDIA Pascal architecture and low computing power); and A2 (entry-level used for edge use cases [1]). As shown in Table 10, GazeVLm consistently improves latency, memory, throughput, and power efficiency. Notably, Vanilla is not runnable on the edge-setting A2 GPU due to memory constraints; however, GazeVLM enables inference, which highlights our feasibility for edge deployment.
>
> Due to OS and system constraints, full deployment on smart glasses and user studies will be addressed as our next piece of work.
>
> [1] NVIDIA, "NVIDIA A2 Tensor Core GPU – Entry-level GPU that brings NVIDIA AI to any server," Available at: https://www.nvidia.com/en-us/data-center/products/a2/ (Accessed Nov 20, 2025).

---

> ### Comment · Reviewer_zVVV · 2025-11-26
>
> Thanks to the authors for the clarification and the additional experiment.
>
> I am still not fully convinced by the cropping experiment. Simply reducing the crop window is not a fair comparison to your method, which has access to the full image anyway. This also raises a conceptual question: if reducing the crop size does not significantly affect accuracy, it implies that most of the informative signal may already come from the global image. A more practical and meaningful baseline would be to use an object detector to crop the object region that the gaze falls on, rather than arbitrarily shrinking the crop window.
>
> Regarding the claim that global and local tokens differ due to resolution or crop size, I remain unconvinced. As long as the underlying context is similar, the tokenized representations should remain largely consistent. This again raises doubts about how much the local tokens are truly contributing. Table 9 also shows only marginal differences, which further suggests that the added value of the local tokens may be limited.
>
> I would also like to add that, although the authors have significantly improved the paper, it still remains unclear to me what the concrete benefits of the proposed approach are compared to a simpler bounding-box-based method. A straightforward baseline that crops regions using detected object boxes—especially those aligned with gaze—seems both practical and competitive. Without stronger empirical evidence or clearer theoretical justification, it is difficult to assess whether the added complexity of the proposed design provides meaningful advantages. I believe the authors are moving in a promising direction, but the method still requires additional justification to convincingly demonstrate its value.

---

> > ### Author Response · Authors · 2025-11-28
> > **Follow-up Response to Reviewer zVVV**
> >
> > **Comparison to object detection-based cropping.** We sincerely thank the reviewer for raising the important question about bounding-box-based cropping as a potential baseline. First of all, we want to clarify that to the best of our knowledge, there is no existing work that uses gaze-based bounding-box cropping for VLM inference. However, we also think that this baseline is a meaningful baseline, so we conducted additional experiments on bounding-box-based cropping.
> >
> > We use YOLO ('yolov8n' model from the `ultralytics` library [2]) and crop the bounding box of the object that has the gaze point. If no object contains the gaze point, we do not crop. We compare with two baselines: (i) bounding-box cropping; and (ii) bounding-box cropping as the local-view image (replacing our local-view image) along with the global-view image.
> >
> > | | Accuracy (%) | Throughput (token/s) | Latency (s) |
> > |:---:|:---:|:---:|:---:|
> > | Vanilla | 74.3 | 1.42 | 8,358 |
> > | Bounding-Box Cropping | 63.2 | 0.71 | 17,471 |
> > | Bounding Box Cropping w/ Global-view | 72.5 | 0.96 | 14,891 |
> > | GazeVLM (1,000 tokens) | 75.2 | 2.18 | 6,242 |
> > | GazeVLM (500 tokens) | 74.8 | 2.70 | 5,274 |
> >
> > As shown in the table above, we have three key findings. First, bounding-box cropping also degrades significantly, since the gaze can be not well-aligned with the correct object, similar to other cropping approaches. This highlights the need of maintaining global coverage instead of relying on a cropped image. Second, bounding-box cropping with the global-view image performs better than the bounding-box cropping-only method; however, it is still worse than our GazeVLM. Our understanding on this is similar to the first reason. Even though we have a global-view image, bounding-box cropping is more sensitive to the quality of gaze points, and thus incorrect local-view cropping results in accuracy degradation. Lastly, the preprocessing time becomes another overhead of object detection; GazeVLM takes 5k-6k seconds of total latency, while bounding-box cropping approaches consume 15k-17k, which highly disturbs the efficiency.
> >
> > We would like to emphasize that our contribution is not simply cropping with gaze but the first gaze-guided token selection framework designed for efficient VLM inference under resource constraints. Existing gaze-aware approaches do not address efficient inference and do not consider robustness to gaze noise.
> >
> > **Contribution of local tokens.** It is important to note that the improvement by introducing dynamic partitioning in LLAVA was 1.4% and 1.8% for GQA and VQAv2 datasets, respectively. Thus, the accuracy gap of 3.6-5.0% between the global-view only and our GazeVLM in Table 9 is not marginal.
> >
> > Additionally, the contribution of the local tokens becomes more important under tighter budgets. As shown in Fig. 9(b), under tight budgets such as 300 tokens, the gap is > 3%, which shows that local tokens meaningfully compensate for missing fine-grained details when the global tokens become coarser.
> >
> > We would be happy to further address any additional questions you may have.
> >
> >
> > [2] Ultralytics, “Object Detection,” Ultralytics YOLO Docs, https://docs.ultralytics.com/tasks/detect/. (Accessed: Nov 27, 2025).

---

### Author Response · Authors · 2025-11-21
**General Response to Reviewers**

We sincerely thank all reviewers for thorough and constructive feedback. Up to this point, we have carefully addressed most of the concerns, newly added substantial results, and verified that all key claims of the paper remain unchanged. The major updates in either the revised manuscript or our responses are summarized as follows:

* We revised repeated text and notations and additionally detailed our token selection strategy by adding Table 3 (summary of notations), Algorithm 2 (token selection), and Fig. 5 (example of selected tokens)

* We added extensive new experiments on gaze-related or user-bias factors: Fig. 11 (leave-one-user-out); Fig. 12(a) (error types); Fig. 12(b) (noise level); Fig. 13 (gaze extraction strategies); Fig. 14(a) (gaze saliency group split); and Fig. 14(b) (gaze saliency criteria)

* We clarified our novelties against baselines (Table 6), added global-only and local-only results (Table 9), and compared our GazeVLM with alternative design choices and adaptive token selection strategies

* We evaluated across various GPUs including edge-level hardware (Table 10) and have started collecting new data using smart glasses

Lastly, we will continue refining the paper, collecting real-world smart-glasses data, and will incorporate experiments on additional models and tasks, as well as further baselines, metrics, and analyses on our best within the rebuttal period.

---

### Meta-Review · Area_Chair_yBQ6 · 2026-01-07

**Summary:**

This paper addresses an important practical problem—improving the efficiency of VLM inference under tight resource constraints—and proposes a gaze-guided, training-free framework with solid empirical engineering. Reviewers generally found the motivation reasonable and the experiments careful, and the rebuttal successfully clarified several implementation details, robustness questions, and baseline comparisons.

However, the reviews consistently raised concerns about limited novelty and the largely heuristic nature of the proposed design, with no strong principled or theoretical grounding. The experimental scope remains narrow, relying on two static VQA datasets, which creates a mismatch with the paper’s motivation around real-time smart-glasses and streaming scenarios. Baseline coverage for efficient VLMs is also limited, and some reviewers remained unconvinced about the necessity and added value of the local-view component over simpler alternatives.

Overall, while the work is promising and well-executed from an engineering perspective, the remaining concerns about novelty, evaluation breadth, and strength of empirical justification prevent the paper from meeting the bar for acceptance at this time.

**Reviewer Concerns:**

Reviewer Concerns

Several reviewer concerns were meaningfully addressed in the rebuttal. The authors substantially improved clarity of the method and implementation details, added missing explanations on token selection and budget enforcement, and responded directly to questions about robustness to gaze noise, user variability, and gaze extraction strategies. Additional ablations, noise injection studies, leave-one-user-out analysis, and detection-based cropping baselines helped resolve many concerns about fairness of comparisons and robustness. At least one reviewer explicitly indicated that these responses addressed their remaining questions.

However, some major concerns remain outstanding. Multiple reviewers continued to question the overall novelty and the largely heuristic nature of the design, noting the lack of a principled or theoretical justification. The evaluation is still limited in scope, with evidence largely restricted to two static VQA datasets, leaving a gap between the real-time smart-glasses motivation and the experimental validation. Baseline coverage for efficient VLMs remains narrow, and some reviewers remained unconvinced about the necessity and added value of the local-view tokens compared to simpler alternatives. These unresolved issues prevent a stronger consensus in favor of acceptance.

**Reviewer Scores:**

Reviewer 48ay
Original: 2
Estimated after discussion: 2
Rationale: Core concerns on novelty, principled design, and real-world validation (smart-glasses, end-to-end deployment) remain largely unaddressed. Added experiments help, but do not resolve the major issues raised. No indication from the reviewer that their concerns were alleviated.

Reviewer 9pQx
Original: 4
Estimated after discussion: 6
Rationale: Reviewer explicitly stated they would consider raising the score if remaining questions were addressed. The authors provided detailed clarifications on baselines, PRE vs. POST logic, gaze extraction, and token accounting, which reasonably addresses the reviewer’s remaining doubts.

Reviewer qxk5
Original: 4
Estimated after discussion: 4
Rationale: While the rebuttal added extensive ablations and analyses, the reviewer’s major concerns on lack of theoretical grounding, limited task/dataset scope, and incomplete baseline coverage remain fundamentally unresolved. No explicit or implicit signal that the score would increase.

Reviewer BFCe
Original: 6
Estimated after discussion: 6
Rationale: The reviewer was already positive. Follow-up questions were answered in detail, but no explicit indication was given that the score would increase beyond the original assessment.

Reviewer zVVV
Original: 4
Estimated after discussion: 4
Rationale: Although the authors added object-detection cropping baselines and further justification, the reviewer explicitly remained unconvinced about the necessity and contribution of local tokens. Major concerns persist.

Score Summary

Estimated scores: 2, 6, 4, 6, 4

Average score: 4.4

---

### Decision · Program_Chairs · 2026-01-26

Reject